# Can Classic GNNs Be Strong Baselines for Graph-level Tasks?
# Simple Architectures Meet Excellence

**Yuankai Luo** [1 2]   **Lei Shi**[* 1]   **Xiao-Ming Wu**[* 2]

## Abstract

Message-passing Graph Neural Networks (GNNs) are often criticized for their limited expressiveness, issues like over-smoothing and over-squashing, and challenges in capturing long-range dependencies. Conversely, Graph Transformers (GTs) are regarded as superior due to their employment of global attention mechanisms, which potentially mitigate these challenges. Literature frequently suggests that GTs outperform GNNs in graph-level tasks, especially for graph classification and regression on small molecular graphs. In this study, we explore the untapped potential of GNNs through an enhanced framework, GNN$^+$, which integrates six widely used techniques: edge feature integration, normalization, dropout, residual connections, feed-forward networks, and positional encoding, to effectively tackle graph-level tasks. We conduct a systematic re-evaluation of three classic GNNs—GCN, GIN, and GatedGCN—enhanced by the GNN$^+$ framework across 14 well-known graph-level datasets. Our results reveal that, contrary to prevailing beliefs, these classic GNNs consistently match or surpass the performance of GTs, securing top-three rankings across all datasets and achieving first place in eight. Furthermore, they demonstrate greater efficiency, running several times faster than GTs on many datasets. This highlights the potential of simple GNN architectures, challenging the notion that complex mechanisms in GTs are essential for superior graph-level performance. Our source code is available at  https://github.com/LUOyk1999/GNNPlus.

[1]Beihang University [2]The Hong Kong Polytechnic University.
[*]Corresponding authors: Lei Shi <{leishi, luoyk}@buaa.edu.cn>, Xiao-Ming Wu <xiao-ming.wu@polyu.edu.hk>.

*Proceedings of the $42^{nd}$ International Conference on Machine Learning*, Vancouver, Canada. PMLR 267, 2025. Copyright 2025 by the author(s).

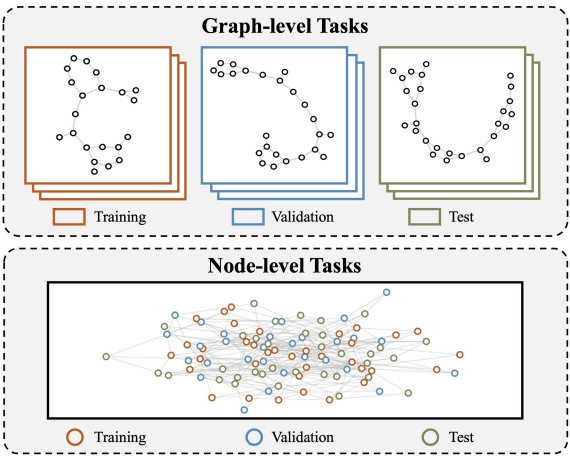

*Figure 1.* Differences between graph-level and node-level tasks.

## 1. Introduction

Graph machine learning addresses both graph-level tasks and node-level tasks, as illustrated in Figure 1. These tasks fundamentally differ in their choice of the basic unit for dataset composition, splitting, and training, with graph-level tasks focusing on the entire graph, while node-level tasks focus on individual nodes. Graph-level tasks (Dwivedi et al., 2023; Hu et al., 2020; Luo et al., 2023b;a) often involve the classification of relatively small molecular graphs in chemistry (Duvenaud et al., 2015; Morris et al., 2020) or the prediction of protein properties in biology (Dwivedi et al., 2022). In contrast, node-level tasks typically involve large social networks (Tang et al., 2009) or citation networks (Yang et al., 2016), where the primary goal is node classification. This distinction in the fundamental unit of dataset leads to differences in methodologies, training strategies, and application domains.

Message-passing Graph Neural Networks (GNNs) (Gilmer et al., 2017), which iteratively aggregate information from local neighborhoods to learn node representations, have become the predominant approach for both graph-level and node-level tasks (Niepert et al., 2016; Kipf & Welling, 2017; Veličković et al., 2018; Xu et al., 2018; Bresson & Laurent, 2017; Wu et al., 2020). Despite their widespread success, GNNs exhibit several inherent limitations, including re-

stricted expressiveness (Xu et al., 2018; Morris et al., 2019), over-smoothing (Li et al., 2018; Chen et al., 2020), over-squashing (Alon & Yahav, 2020), and a limited capacity to capture long-range dependencies (Dwivedi et al., 2022).

A prevalent perspective is that Graph Transformers (GTs) (Müller et al., 2023; Min et al., 2022; Luo et al., 2024c;a), as an alternative to GNNs, leverage global attention mechanisms that enable each node to attend to all others (Yun et al., 2019; Dwivedi & Bresson, 2020), effectively modeling long-range interactions and addressing issues such as over-smoothing, over-squashing, and limited expressiveness (Rampášek et al., 2022; Zhang et al., 2023). Consequently, GTs have become the preferred method for graph-level tasks, especially in the context of small molecular graphs (Ma et al., 2023). However, the quadratic complexity of global attention mechanisms limits the scalability of GTs in large-scale, real-world applications (Behrouz & Hashemi, 2024; Sancak et al., 2024; Ding et al., 2024). Moreover, it has been noted that many state-of-the-art GTs (Chen et al., 2022; Rampášek et al., 2022; Shirzad et al., 2023; Ma et al., 2023) still rely—either explicitly or implicitly—on the message passing mechanism of GNNs to learn local node representations, thereby enhancing performance.

Recent studies (Luo et al., 2024b; 2025a;b) have demonstrated that classic GNNs such as GCN (Kipf & Welling, 2017), GAT (Veličković et al., 2018), and Graph-SAGE (Hamilton et al., 2017) can achieve performance comparable to, or even exceeding, that of state-of-the-art GTs for node-level tasks. However, a similar conclusion has not yet been established for graph-level tasks. While Tönshoff et al. (2023) conducted pioneering research demonstrating that tuning a few hyperparameters can significantly enhance the performance of classic GNNs, their results indicate that these models still do not match the overall performance of GTs. Furthermore, their investigation is limited to the Long-Range Graph Benchmark (LRGB) (Dwivedi et al., 2022). This raises an important question: *"Can classic GNNs also excel in graph-level tasks?"*

To thoroughly investigate this question, we introduce GNN$^+$, an enhanced GNN framework that incorporates established techniques into the message-passing mechanism, to effectively address graph-level tasks. As illustrated in Fig. 2, GNN$^+$ integrates six widely used techniques: the incorporation of edge features (Gilmer et al., 2017), normalization (Ioffe & Szegedy, 2015), dropout (Srivastava et al., 2014), residual connections (He et al., 2016), feed-forward networks (FFN) (Vaswani et al., 2017), and positional encoding (Vaswani et al., 2017). Each technique serves as a hyperparameter that can be tuned to optimize performance.

We systematically re-evaluate 3 classic GNNs—GCN (Kipf & Welling, 2017), GIN (Xu et al., 2018), and Gat-edGCN (Bresson & Laurent, 2017)—enhanced by the

GNN$^+$ framework across 14 well-known graph-level datasets from GNN Benchmark (Dwivedi et al., 2023), LRGB (Dwivedi et al., 2022), and OGB (Hu et al., 2020). The results show that the enhanced versions of classic GNNs match or even outperform state-of-the-art (SOTA) GTs, consistently securing **top-three** rankings and achieving **first place in eight datasets**. Moreover, they exhibit superior efficiency, running several times faster than GTs on many datasets. These findings provide a *positive answer* to the previously posed question, suggesting that the true potential of GNNs for graph-level applications has been underestimated. The GNN$^+$ framework effectively unlocks this potential while addressing inherent limitations.

Furthermore, in our ablation study, we highlight the importance of each technique integrated into GNN$^+$. Specifically,

(1) **Edge features** are particularly beneficial in molecular and image superpixel datasets, where they encode crucial domain-specific information.

(2) **Normalization** becomes more crucial as the scale of datasets increases.

(3) **Dropout** benefits most graph-level datasets, where a modest dropout rate proves both sufficient and optimal.

(4) **Residual connections** are consistently essential, except in shallow GNNs applied to small graphs.

(5) **FFN** is especially pivotal for simpler models, such as GCN, in graph-level tasks.

(6) **Positional encoding** plays a more critical role in small-scale datasets compared to large-scale ones.

These findings underscore the nuanced roles each technique plays in enhancing the performance of classic GNNs, guiding future explorations in GNN design and application.

## 2. Classic GNNs for Graph-level Tasks

Define a graph as $\mathcal{G} = (\mathcal{V}, \mathcal{E}, \boldsymbol{X}, \boldsymbol{E})$, where $\mathcal{V}$ is the set of nodes, and $\mathcal{E} \subseteq \mathcal{V} \times \mathcal{V}$ is the set of edges. The node feature matrix is $\boldsymbol{X} \in \mathbb{R}^{|\mathcal{V}| \times d_{\mathcal{V}}}$, where $|\mathcal{V}|$ is the number of nodes, and $d_{\mathcal{V}}$ is the dimension of the node features. The edge feature matrix is $\boldsymbol{E} \in \mathbb{R}^{|\mathcal{E}| \times d_{\mathcal{E}}}$, where $|\mathcal{E}|$ is the number of edges and $d_{\mathcal{E}}$ is the dimension of the edge features. Let $\boldsymbol{A} \in \mathbb{R}^{|\mathcal{V}| \times |\mathcal{V}|}$ denote the adjacency matrix of $\mathcal{G}$.

**Message-passing Graph Neural Networks (GNNs)** compute node representations $\boldsymbol{h}_v^l$ at each layer $l$ via a message-passing mechanism, defined by Gilmer et al. (2017):

$$\boldsymbol{h}_v^l = \text{UPDATE}^l \left( \boldsymbol{h}_v^{l-1}, \text{AGG}^l \left( \left\{ \boldsymbol{h}_u^{l-1} \mid u \in \mathcal{N}(v) \right\} \right) \right),$$
(1)

where $\mathcal{N}(v)$ represents the neighboring nodes adjacent to $v$, $\text{AGG}^l$ is the message aggregation function, and $\text{UPDATE}^l$ is the update function. Initially, each node $v$ is assigned a

feature vector $\boldsymbol{h}_v^0 = \boldsymbol{x}_v \in \mathbb{R}^d$. The function $\text{AGG}^l$ is then used to aggregate information from the neighbors of $v$ to update its representation. The output of the last layer $L$, i.e., $\text{GNN}(v, \boldsymbol{A}, \boldsymbol{X}) = \boldsymbol{h}_v^L$, is the representation of $v$ produced by the GNN. In this work, we focus on three classic GNNs: GCN (Kipf & Welling, 2017), GIN (Xu et al., 2018), and GatedGCN (Bresson & Laurent, 2017), which differ in their approach to learning the node representation $\boldsymbol{h}_v^l$.

**Graph Convolutional Networks (GCN)** (Kipf & Welling, 2017), the vanilla GCN model, is formulated as:

$$\boldsymbol{h}_v^l = \sigma\left(\sum_{u \in \mathcal{N}(v) \cup \{v\}} \frac{1}{\sqrt{\hat{d}_u \hat{d}_v}} \boldsymbol{h}_u^{l-1} \boldsymbol{W}^l\right), \qquad (2)$$

where $\hat{d}_v = 1 + \sum_{u \in \mathcal{N}(v)} 1$, $\sum_{u \in \mathcal{N}(v)} 1$ denotes the degree of node $v$, $\boldsymbol{W}^l$ is the trainable weight matrix in layer $l$, and $\sigma$ is the activation function, e.g., $\text{ReLU}(\cdot) = \max(0, \cdot)$.

**Graph Isomorphism Networks (GIN)** (Xu et al., 2018) learn node representations through a different approach:

$$\boldsymbol{h}_v^l = \text{MLP}^l\left((1 + \epsilon) \cdot \boldsymbol{h}_v^{l-1} + \sum_{u \in \mathcal{N}(v)} \boldsymbol{h}_u^{l-1}\right), \qquad (3)$$

where $\epsilon$ is a constant, typically set to 0, and $\text{MLP}^l$ denotes a multi-layer perceptron, which usually consists of 2 layers.

**Residual Gated Graph Convolutional Networks (GatedGCN)** (Bresson & Laurent, 2017) enhance traditional graph convolutions by incorporating gating mechanisms, improving adaptability and expressiveness:

$$\boldsymbol{h}_v^l = \boldsymbol{h}_v^{l-1} \boldsymbol{W}_1^l + \sum_{u \in \mathcal{N}(v)} \boldsymbol{\eta}_{v,u} \odot \boldsymbol{h}_u^{l-1} \boldsymbol{W}_2^l, \qquad (4)$$

where $\boldsymbol{\eta}_{v,u} = \sigma(\boldsymbol{h}_v^{l-1} \boldsymbol{W}_3^l + \boldsymbol{h}_u^{l-1} \boldsymbol{W}_4^l)$ is the gating function, and $\sigma$ denotes the sigmoid activation function. This gating function determines how much each neighboring node contributes to updating the representation of the current node. The matrices $\boldsymbol{W}_1^l$, $\boldsymbol{W}_2^l$, $\boldsymbol{W}_3^l$, $\boldsymbol{W}_4^l$ are trainable weight matrices specific to the layer $l$.

**Graph-level tasks** treat the entire graph, rather than individual nodes or edges, as the fundamental unit for dataset composition, splitting, and training. Formally, given a labeled graph dataset $\Gamma = \{(\mathcal{G}_i, \boldsymbol{y}_i)\}_{i=1}^n$, each graph $\mathcal{G}_i$ is associated with a label vector $\boldsymbol{y}_i$, representing either categorical labels for classification or continuous values for regression. Next, the dataset $\Gamma$ is typically split into training, validation, and test sets, denoted as $\Gamma = \Gamma_{\text{train}} \cup \Gamma_{\text{val}} \cup \Gamma_{\text{test}}$.

Graph-level tasks encompass inductive prediction tasks that operate on entire graphs, as well as on individual nodes or edges (Dwivedi et al., 2022), with each corresponding to a distinct label vector $\boldsymbol{y}_i$. Each type of task requires a tailored graph readout function R, which aggregates the output

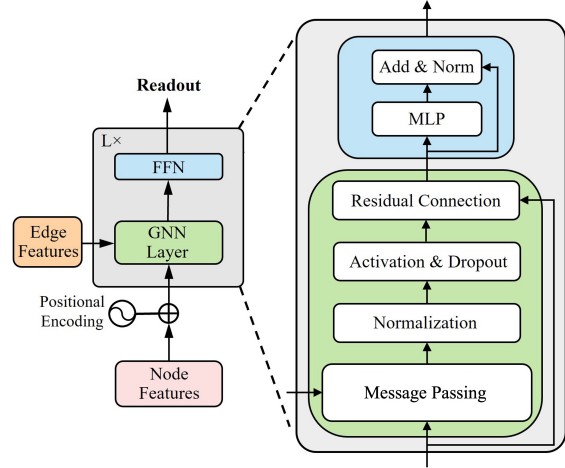

*Figure 2.* The architecture of GNN$^+$.

representations to compute the readout result, expressed as:

$$\boldsymbol{h}_i^{\text{readout}} = \text{R}\left(\left\{\boldsymbol{h}_v^L : v \in \mathcal{V}_i\right\}\right), \qquad (5)$$

where $\mathcal{V}_i$ represents the set of nodes in the graph $\mathcal{G}_i$. For example, for *graph prediction tasks*, which aim to make predictions about the entire graph, the readout function R often operates as a global mean pooling function.

Finally, for any graph $\mathcal{G}_i$, the readout result is passed through a prediction head $g(\cdot)$ to obtain the predicted label $\hat{\boldsymbol{y}}_i = g(\boldsymbol{h}_i^{\text{readout}})$. The training objective is to minimize the total loss $L(\boldsymbol{\theta}) = \sum_{\mathcal{G}_i \in \Gamma_{\text{train}}} \ell(\hat{\boldsymbol{y}}_i, \boldsymbol{y}_i)$ w.r.t. all graphs in the training set $\Gamma_{\text{train}}$, where $\boldsymbol{y}_i$ represents the ground-truth label of $\mathcal{G}_i$ and $\boldsymbol{\theta}$ denotes the trainable GNN parameters.

## 3. GNN$^+$: Enhancing Classic GNNs for Graph-level Tasks

We propose an enhancement to classic GNNs for graph-level tasks by incorporating six popular techniques: edge feature integration, normalization, dropout, residual connections, feed-forward networks (FFN), and positional encoding. The enhanced framework, GNN$^+$, is illustrated in Figure 2.

### 3.1. Edge Feature Integration

Edge features were initially incorporated into some GNN frameworks (Gilmer et al., 2017; Hu et al., 2019) by directly integrating them into the message-passing process to enhance information propagation between nodes. Following this practice, GraphGPS (Rampášek et al., 2022) and subsequent GTs encode edge features within their local modules to enrich node representations.

Taking GCN (Eq. 2) as an example, the edge features are

integrated into the massage-passing process as follows:

$$h_v^l = \sigma\left(\sum_{u \in \mathcal{N}(v) \cup \{v\}} \frac{1}{\sqrt{\hat{d}_u \hat{d}_v}} h_u^{l-1} W^l + e_{uv} W_e^l\right), \quad (6)$$

where $W_e^l$ is the trainable weight matrix in layer $l$, and $e_{uv}$ is the feature vector of the edge between $u$ and $v$.

### 3.2. Normalization

Normalization techniques play a critical role in stabilizing the training of GNNs by mitigating the effects of *covariate shift*, where the distribution of node embeddings changes across layers during training. By normalizing node embeddings at each layer, the training process becomes more stable, enabling the use of higher learning rates and achieving faster convergence (Cai et al., 2021).

Batch Normalization (BN) (Ioffe & Szegedy, 2015) and Layer Normalization (LN) (Ba et al., 2016) are widely used techniques, typically applied to the output of each layer *before* the activation function $\sigma(\cdot)$. Here, we use BN:

$$h_v^l = \sigma\left(\text{BN}\left(\sum_{u \in \mathcal{N}(v) \cup \{v\}} \frac{1}{\sqrt{\hat{d}_u \hat{d}_v}} h_u^{l-1} W^l + e_{uv} W_e^l\right)\right). \quad (7)$$

### 3.3. Dropout

Dropout (Srivastava et al., 2014), a technique widely used in convolutional neural networks (CNNs) to address overfitting by reducing co-adaptation among hidden neurons (Hinton et al., 2012; Yosinski et al., 2014), has also been found to be effective in addressing similar issues in GNNs (Shu et al., 2022), where the co-adaptation effects propagate and accumulate via message passing among different nodes. Typically, dropout is applied to the embeddings *after* activation:

$$h_v^l = \text{Dropout}\left(\sigma\left(\text{BN}\left(\sum_{u \in \mathcal{N}(v) \cup \{v\}} \frac{1}{\sqrt{\hat{d}_u \hat{d}_v}} h_u^{l-1} W^l \right.\right.\right.$$
$$\left.\left.\left. + e_{uv} W_e^l\right)\right)\right). \quad (8)$$

### 3.4. Residual Connection

Residual connections (He et al., 2016) significantly enhance CNN performance by directly connecting the input of a layer to its output, thus alleviating the problem of vanishing gradient. They were first adopted by the vanilla GCN (Kipf & Welling, 2017) and has since been incorporated into subsequent works such as GatedGCN (Bresson & Laurent, 2017) and DeepGCNs (Li et al., 2019). Formally, residual connections can be integrated into GNNs as follows:

$$h_v^l = \text{Dropout}\left(\sigma\left(\text{BN}\left(\sum_{u \in \mathcal{N}(v) \cup \{v\}} \frac{1}{\sqrt{\hat{d}_u \hat{d}_v}} h_u^{l-1} W^l \right.\right.\right.$$
$$\left.\left.\left. + e_{uv} W_e^l\right)\right)\right) + h_v^{l-1}. \quad (9)$$

*Table 1.* Overview of the datasets used for graph-level tasks.

| Dataset | # graphs | Avg. # nodes | Avg. # edges | Task Type |
|---|---|---|---|---|
| ZINC | 12,000 | 23.2 | 24.9 | Graph regression |
| MNIST | 70,000 | 70.6 | 564.5 | Graph classification |
| CIFAR10 | 60,000 | 117.6 | 941.1 | Graph classification |
| PATTERN | 14,000 | 118.9 | 3,039.3 | Inductive node cls. |
| CLUSTER | 12,000 | 117.2 | 2,150.9 | Inductive node cls. |
| Peptides-func | 15,535 | 150.9 | 307.3 | Graph classification |
| Peptides-struct | 15,535 | 150.9 | 307.3 | Graph regression |
| PascalVOC-SP | 11,355 | 479.4 | 2,710.5 | Inductive node cls. |
| COCO-SP | 123,286 | 476.9 | 2,693.7 | Inductive node cls. |
| MalNet-Tiny | 5,000 | 1,410.3 | 2,859.9 | Graph classification |
| ogbg-molhiv | 41,127 | 25.5 | 27.5 | Graph classification |
| ogbg-molpcba | 437,929 | 26.0 | 28.1 | Graph classification |
| ogbg-ppa | 158,100 | 243.4 | 2,266.1 | Graph classification |
| ogbg-code2 | 452,741 | 125.2 | 124.2 | Graph classification |

While deeper networks, such as deep CNNs (He et al., 2016; Huang et al., 2017), are capable of extract more complex features, GNNs encounter challenges like over-smoothing (Li et al., 2018), where deeper models lead to indistinguishable node representations. Consequently, most GNNs are shallow, typically with 2 to 5 layers. However, by incorporating residual connections, we show that deeper GNNs, ranging from 3 to 20 layers, can achieve strong performance.

### 3.5. Feed-Forward Network

GTs incorporate a feed-forward network (FFN) as a crucial component within each of their layers. The FFN enhances the model's ability to perform complex feature transformations and introduces non-linearity, thereby increasing the network's expressive power. Inspired by this, we propose appending a fully-connected FFN at the end of each layer of GNNs, defined as:

$$\text{FFN}(h) = \text{BN}(\sigma(h W_{\text{FFN}_1}^l) W_{\text{FFN}_2}^l + h), \quad (10)$$

where $W_{\text{FFN}_1}^l$ and $W_{\text{FFN}_2}^l$ are the trainable weight matrices of the FFN at the $l$-th GNN layer. The node embeddings output by the FFN are then computed as:

$$h_v^l = \text{FFN}\left(\text{Dropout}\left(\sigma\left(\text{BN}\left(\sum_{u \in \mathcal{N}(v) \cup \{v\}} \frac{1}{\sqrt{\hat{d}_u \hat{d}_v}} h_u^{l-1} W^l \right.\right.\right.\right.$$
$$\left.\left.\left.\left. + e_{uv} W_e^l\right)\right)\right) + h_v^{l-1}\right). \quad (11)$$

### 3.6. Positional Encoding

Positional encoding (PE) was introduced in the Transformer model (Vaswani et al., 2017) to represent the positions of tokens within a sequence for language modeling. In GTs, PE is used to incorporate graph positional or structural information. The encodings are typically added or concatenated to the input node features $x_v$ before being fed into the GTs. Various PE methods have been proposed, such as Laplacian Positional Encoding (LapPE) (Dwivedi & Bresson, 2020;

*Table 2.* Test performance on five benchmarks from (Dwivedi et al., 2023) (%). Shown is the mean $\pm$ s.d. of 5 runs with different random seeds. $^+$ denotes the enhanced version, while the baseline results were obtained from their respective original papers. # Param $\sim 500$K for ZINC, PATTERN, and CLUSTER, and $\sim 100$K for MNIST and CIFAR10. The top $1^{st}$, $2^{nd}$ and $3^{rd}$ results are highlighted.

| | ZINC | MNIST | CIFAR10 | PATTERN | CLUSTER |
|---|---|---|---|---|---|
| # graphs | 12,000 | 70,000 | 60,000 | 14,000 | 12,000 |
| Avg. # nodes | 23.2 | 70.6 | 117.6 | 118.9 | 117.2 |
| Avg. # edges | 24.9 | 564.5 | 941.1 | 3039.3 | 2150.9 |
| Metric | MAE $\downarrow$ | Accuracy $\uparrow$ | Accuracy $\uparrow$ | Accuracy $\uparrow$ | Accuracy $\uparrow$ |
| GT (2020) | $0.226 \pm 0.014$ | $90.831 \pm 0.161$ | $59.753 \pm 0.293$ | $84.808 \pm 0.068$ | $73.169 \pm 0.622$ |
| SAN (2021) | $0.139 \pm 0.006$ | – | – | $86.581 \pm 0.037$ | $76.691 \pm 0.650$ |
| Graphormer (2021) | $0.122 \pm 0.006$ | – | – | – | – |
| SAT (2022) | $0.094 \pm 0.008$ | – | – | $86.848 \pm 0.037$ | $77.856 \pm 0.104$ |
| EGT (2022) | $0.108 \pm 0.009$ | $98.173 \pm 0.087$ | $68.702 \pm 0.409$ | $86.821 \pm 0.020$ | $79.232 \pm 0.348$ |
| GraphGPS (2022) | $0.070 \pm 0.004$ | $98.051 \pm 0.126$ | $72.298 \pm 0.356$ | $86.685 \pm 0.059$ | $78.016 \pm 0.180$ |
| GRPE (2022) | $0.094 \pm 0.002$ | – | – | $87.020 \pm 0.042$ | – |
| Graphormer-URPE (2022) | $0.086 \pm 0.007$ | – | – | – | – |
| Graphormer-GD (2023) | $0.081 \pm 0.009$ | – | – | – | – |
| Specformer (2023) | $0.066 \pm 0.003$ | – | – | – | – |
| LGI-GT (2023) | – | – | – | $86.930 \pm 0.040$ | – |
| GPTrans-Nano (2023b) | – | – | – | $86.731 \pm 0.085$ | – |
| Graph ViT/MLP-Mixer (2023) | $0.073 \pm 0.001$ | $98.460 \pm 0.090$ | $73.960 \pm 0.330$ | – | – |
| Exphormer (2023) | – | $98.414 \pm 0.038$ | $74.754 \pm 0.194$ | $86.734 \pm 0.008$ | – |
| GRIT (2023) | $0.059 \pm 0.002$ | $98.108 \pm 0.111$ | $76.468 \pm 0.881$ | $87.196 \pm 0.076$ | $80.026 \pm 0.277$ |
| GRED (2024) | $0.077 \pm 0.002$ | $98.383 \pm 0.012$ | $76.853 \pm 0.185$ | $86.759 \pm 0.020$ | $78.495 \pm 0.103$ |
| GEAET (2024) | – | $98.513 \pm 0.086$ | $76.634 \pm 0.427$ | $86.993 \pm 0.026$ | – |
| TIGT (2024) | $0.057 \pm 0.002$ | $98.231 \pm 0.132$ | $73.963 \pm 0.361$ | $86.681 \pm 0.062$ | $78.025 \pm 0.223$ |
| Cluster-GT (2024a) | $0.071 \pm 0.004$ | – | – | – | – |
| GMN (2024) | – | $98.391 \pm 0.182$ | $74.560 \pm 0.381$ | $87.090 \pm 1.260$ | – |
| Graph-Mamba (2024) | – | $98.420 \pm 0.080$ | $73.700 \pm 0.340$ | $86.710 \pm 0.050$ | $76.800 \pm 0.360$ |
| GCN | $0.367 \pm 0.011$ | $90.705 \pm 0.218$ | $55.710 \pm 0.381$ | $71.892 \pm 0.334$ | $68.498 \pm 0.976$ |
| **GCN$^+$** | $0.076 \pm 0.009$ **79.3%$\downarrow$** | $98.382 \pm 0.095$ **8.5%$\uparrow$** | $69.824 \pm 0.413$ **25.4%$\uparrow$** | $87.021 \pm 0.095$ **21.1%$\uparrow$** | $77.109 \pm 0.872$ **12.6%$\uparrow$** |
| GIN | $0.526 \pm 0.051$ | $96.485 \pm 0.252$ | $55.255 \pm 1.527$ | $85.387 \pm 0.136$ | $64.716 \pm 1.553$ |
| **GIN$^+$** | $0.065 \pm 0.004$ **87.6%$\downarrow$** | $98.285 \pm 0.103$ **1.9%$\uparrow$** | $69.592 \pm 0.287$ **25.9%$\uparrow$** | $86.842 \pm 0.048$ **1.7%$\uparrow$** | $74.794 \pm 0.213$ **15.6%$\uparrow$** |
| GatedGCN | $0.282 \pm 0.015$ | $97.340 \pm 0.143$ | $67.312 \pm 0.311$ | $85.568 \pm 0.088$ | $73.840 \pm 0.326$ |
| **GatedGCN$^+$** | $0.077 \pm 0.005$ **72.7%$\downarrow$** | $98.712 \pm 0.137$ **1.4%$\uparrow$** | $77.218 \pm 0.381$ **14.7%$\uparrow$** | $87.029 \pm 0.037$ **1.7%$\uparrow$** | $79.128 \pm 0.235$ **7.1%$\uparrow$** |
| Time (epoch) of GraphGPS | 21s | 76s | 64s | 32s | 86s |
| Time (epoch) of **GCN$^+$** | **7s** | **60s** | **40s** | **19s** | **29s** |

Kreuzer et al., 2021), Weisfeiler-Lehman Positional Encoding (WLPE) (Zhang et al., 2020), Random Walk Structural Encoding (RWSE) (Li et al., 2020; Dwivedi et al., 2021; Rampášek et al., 2022), Learnable Structural and Positional Encodings (LSPE) (Dwivedi et al., 2021), and Relative Random Walk Probabilities (RRWP) (Ma et al., 2023). Following the practice, we use RWSE, one of the most efficient PE methods, to improve the performance of GNNs as follows:

$$\boldsymbol{x}_v = [\boldsymbol{x}_v \| \boldsymbol{x}_v^{\text{RWSE}}] \boldsymbol{W}_{\text{PE}}, \qquad (12)$$

where $[\cdot \| \cdot]$ denotes concatenation, $\boldsymbol{x}_v^{\text{RWSE}}$ represents the RWSE of node $v$, and $\boldsymbol{W}_{\text{PE}}$ is the trainable weight matrix.

## 4. Assessment: Experimental Setup

**Datasets, Table 1.** We use widely adopted graph-level datasets in our experiments, including **ZINC**, **MNIST**, **CIFAR10**, **PATTERN**, and **CLUSTER** from the GNN Benchmark (Dwivedi et al., 2023); **Peptides-func**, **Peptides-struct**, **PascalVOC-SP**, **COCO-SP**, and **MalNet-Tiny** from Long-Range Graph Benchmark (LRGB) (Dwivedi et al., 2022; Freitas & Dong, 2021); and **ogbg-molhiv**, **ogbg-**

**molpcba**, **ogbg-ppa**, and **ogbg-code2** from Open Graph Benchmark (OGB) (Hu et al., 2020). We follow their respective standard evaluation protocols including the splits and metrics. For further details, refer to the Appendix A.2.

**Baselines.** Our main focus lies on classic GNNs: **GCN** (Kipf & Welling, 2017), **GIN** (Xu et al., 2018; Hu et al., 2019), **GatedGCN** (Bresson & Laurent, 2017), the SOTA GTs: GT (2020), GraphTrans (2021), SAN (2021), Graphormer (2021), SAT (2022), EGT (2022), GraphGPS (2022; 2023), GRPE (2022), Graphormer-URPE (2022), Graphormer-GD (2023), Specformer (2023), LGI-GT (2023), GPTrans-Nano (2023b), Graph ViT/MLP-Mixer (2023), NAGphormer (2023a), DIFFormer (2023), MGT (2023), DRew (2023), Exphormer (2023), GRIT (2023), GRED (2024), GEAET (2024), Subgraphormer (2024), TIGT (2024), GECO (2024), GPNN (2024), Cluster-GT (2024a), and the SOTA graph state space models (GSSMs): GMN (2024), Graph-Mamba (2024), GSSC (2024b). Furthermore, various other GTs exist in related surveys (Hoang et al., 2024; Shehzad et al., 2024; Müller et al., 2023), empirically shown to be inferior to the GTs we compared against for graph-level tasks. We report the performance results of

*Table 3.* Test performance on five datasets from Long-Range Graph Benchmarks (LRGB) (Dwivedi et al., 2022; Freitas & Dong, 2021). $^+$ denotes the enhanced version, while the baseline results were obtained from their respective original papers. # Param $\sim$ 500K for all.

| | Peptides-func | Peptides-struct | PascalVOC-SP | COCO-SP | MalNet-Tiny |
|---|---|---|---|---|---|
| # graphs | 15,535 | 15,535 | 11,355 | 123,286 | 5,000 |
| Avg. # nodes | 150.9 | 150.9 | 479.4 | 476.9 | 1,410.3 |
| Avg. # edges | 307.3 | 307.3 | 2,710.5 | 2,693.7 | 2,859.9 |
| Metric | Avg. Precision ↑ | MAE ↓ | F1 score ↑ | F1 score ↑ | Accuracy ↑ |
| GT (2020) | $0.6326 \pm 0.0126$ | $0.2529 \pm 0.0016$ | $0.2694 \pm 0.0098$ | $0.2618 \pm 0.0031$ | – |
| SAN (2021) | $0.6439 \pm 0.0075$ | $0.2545 \pm 0.0012$ | $0.3230 \pm 0.0039$ | $0.2592 \pm 0.0158$ | – |
| GraphGPS (2022) | $0.6535 \pm 0.0041$ | $0.2500 \pm 0.0005$ | $0.3748 \pm 0.0109$ | $0.3412 \pm 0.0044$ | $0.9350 \pm 0.0041$ |
| GraphGPS (2023) | $0.6534 \pm 0.0091$ | $0.2509 \pm 0.0014$ | $0.4440 \pm 0.0065$ | $0.3884 \pm 0.0055$ | $0.9350 \pm 0.0041$ |
| NAGphormer (2023a) | – | – | $0.4006 \pm 0.0061$ | $0.3458 \pm 0.0070$ | – |
| DIFFormer (2023) | – | – | $0.3988 \pm 0.0045$ | $0.3620 \pm 0.0012$ | – |
| MGT (2023) | $0.6817 \pm 0.0064$ | $0.2453 \pm 0.0025$ | – | – | – |
| DRew (2023) | $0.7150 \pm 0.0044$ | $0.2536 \pm 0.0015$ | $0.3314 \pm 0.0024$ | – | – |
| Graph ViT/MLP-Mixer (2023) | $0.6970 \pm 0.0080$ | $0.2449 \pm 0.0016$ | – | – | – |
| Exphormer (2023) | $0.6258 \pm 0.0092$ | $0.2512 \pm 0.0025$ | $0.3446 \pm 0.0064$ | $0.3430 \pm 0.0108$ | $0.9402 \pm 0.0021$ |
| GRIT (2023) | $0.6988 \pm 0.0082$ | $0.2460 \pm 0.0012$ | – | – | – |
| Subgraphormer (2024) | $0.6415 \pm 0.0052$ | $0.2475 \pm 0.0007$ | – | – | – |
| GRED (2024) | $0.7133 \pm 0.0011$ | $0.2455 \pm 0.0013$ | – | – | – |
| GEAET (2024) | $0.6485 \pm 0.0035$ | $0.2547 \pm 0.0009$ | $0.3933 \pm 0.0027$ | $0.3219 \pm 0.0052$ | – |
| TIGT (2024) | $0.6679 \pm 0.0074$ | $0.2485 \pm 0.0015$ | – | – | – |
| GECO (2024) | $0.6975 \pm 0.0025$ | $0.2464 \pm 0.0009$ | $0.4210 \pm 0.0080$ | $0.3320 \pm 0.0032$ | – |
| GPNN (2024) | $0.6955 \pm 0.0057$ | $0.2454 \pm 0.0003$ | – | – | – |
| Graph-Mamba (2024) | $0.6739 \pm 0.0087$ | $0.2478 \pm 0.0016$ | $0.4191 \pm 0.0126$ | $0.3960 \pm 0.0175$ | $0.9340 \pm 0.0027$ |
| GSSC (2024b) | $0.7081 \pm 0.0062$ | $0.2459 \pm 0.0020$ | $0.4561 \pm 0.0039$ | – | $0.9406 \pm 0.0064$ |
| GCN | $0.6860 \pm 0.0050$ | $0.2460 \pm 0.0007$ | $0.2078 \pm 0.0031$ | $0.1338 \pm 0.0007$ | $0.8100 \pm 0.0081$ |
| **GCN$^+$** | $0.7261 \pm 0.0067$ **5.9%↑** | $0.2421 \pm 0.0016$ **1.6%↓** | $0.3357 \pm 0.0087$ **62.0%↑** | $0.2733 \pm 0.0041$ **104.9%↑** | $0.9354 \pm 0.0045$ **15.5%↑** |
| GIN | $0.6621 \pm 0.0067$ | $0.2473 \pm 0.0017$ | $0.2718 \pm 0.0054$ | $0.2125 \pm 0.0009$ | $0.8898 \pm 0.0055$ |
| **GIN$^+$** | $0.7059 \pm 0.0089$ **6.6%↑** | $0.2429 \pm 0.0019$ **1.8%↓** | $0.3189 \pm 0.0105$ **17.3%↑** | $0.2483 \pm 0.0046$ **16.9%↑** | $0.9325 \pm 0.0040$ **4.8%↑** |
| GatedGCN | $0.6765 \pm 0.0047$ | $0.2477 \pm 0.0009$ | $0.3880 \pm 0.0040$ | $0.2922 \pm 0.0018$ | $0.9223 \pm 0.0065$ |
| **GatedGCN$^+$** | $0.7006 \pm 0.0033$ **3.6%↑** | $0.2431 \pm 0.0020$ **1.9%↓** | $0.4263 \pm 0.0057$ **9.9%↑** | $0.3802 \pm 0.0015$ **30.1%↑** | $0.9460 \pm 0.0057$ **2.6%↑** |
| Time (epoch) of GraphGPS | 6s | 6s | 17s | 213s | 46s |
| Time (epoch) of **GCN$^+$** | 6s | 6s | **12s** | **162s** | **6s** |

baselines primarily from (Rampášek et al., 2022; Tönshoff et al., 2023; Ma et al., 2023; Wang et al., 2024), with the remaining obtained from their respective original papers or official leaderboards whenever possible, as those results are obtained by well-tuned models.

**Hyperparameter Configurations.** We conduct hyperparameter tuning on 3 classic GNNs, consistent with the hyperparameter search space of GraphGPS (Rampášek et al., 2022; Tönshoff et al., 2023). Specifically, we utilize the AdamW optimizer (Loshchilov, 2017) with a learning rate from $\{0.0001, 0.0005, 0.001\}$ and an epoch limit of 2000. As discussed in Section 3, we focus on whether to use the edge feature module, normalization (BN), residual connections, FFN, PE (RWSE), and dropout rates from $\{0.05, 0.1, 0.15, 0.2, 0.3\}$, the number of layers from 3 to 20. Considering the large number of hyperparameters and datasets, we do not perform an exhaustive search. Additionally, *we retrain baseline GTs using the same hyperparameter search space and training environments as the classic GNNs. Since the retrained results did not surpass those in their original papers, we present the results from those sources.* **GNN$^+$** denotes the enhanced version. We report mean scores and standard deviations after 5 independent runs with different random seeds. Detailed hyperparameters are provided in Appendix A.

## 5. Assessment: Results and Findings

### 5.1. Overall Performance

We evaluate the performance of the enhanced versions of 3 classic GNNs across 14 well-known graph-level datasets.

> The enhanced versions of classic GNNs achieved state-of-the-art performance, ranking in the **top three across 14 datasets**, including **first place in 8 of them**, while also demonstrating **superior efficiency**. This suggests that the GNN$^+$ framework effectively harnesses the potential of classic GNNs for graph-level tasks and successfully mitigates their inherent limitations.

**GNN Benchmark, Table 2.** We observe that our GNN$^+$ implementation substantially enhances the performance of classic GNNs, with the most significant improvements on ZINC, PATTERN, and CLUSTER. On MNIST and CIFAR, GatedGCN$^+$ outperforms SOTA models such as GEAET and GRED, securing top rankings.

**Long-Range Graph Benchmark (LRGB), Table 3.** The results reveal that classic GNNs can achieve strong performance across LRGB datasets. Specifically, GCN$^+$ excels on the Peptides-func and Peptides-struct datasets. On the other hand, GatedGCN$^+$ achieves the highest accuracy on

*Table 4.* Test performance in four benchmarks from Open Graph Benchmark (OGB) (Hu et al., 2020). $^+$ denotes the enhanced version, while the baseline results were obtained from their respective original papers. $^\dagger$ indicates the use of additional pretraining datasets, included here for reference only and excluded from ranking.

| | ogbg-molhiv | ogbg-molpcba | ogbg-ppa | ogbg-code2 |
|---|---|---|---|---|
| # graphs | 41,127 | 437,929 | 158,100 | 452,741 |
| Avg. # nodes | 25.5 | 26.0 | 243.4 | 125.2 |
| Avg. # edges | 27.5 | 28.1 | 2,266.1 | 124.2 |
| Metric | AUROC ↑ | Avg. Precision ↑ | Accuracy ↑ | F1 score ↑ |
| GT (2020) | – | – | $0.6454_{\pm 0.0033}$ | $0.1670_{\pm 0.0015}$ |
| GraphTrans (2021) | – | $0.2761_{\pm 0.0029}$ | – | $0.1830_{\pm 0.0024}$ |
| SAN (2021) | $0.7785_{\pm 0.2470}$ | $0.2765_{\pm 0.0042}$ | – | – |
| Graphormer (pre-trained) (2021) | $0.8051_{\pm 0.0053}^\dagger$ | – | – | – |
| SAT (2022) | – | – | $0.7522_{\pm 0.0056}$ | $\mathbf{0.1937}_{\pm 0.0028}$ |
| EGT (pre-trained) (2022) | $0.8060_{\pm 0.0065}^\dagger$ | $0.2961_{\pm 0.0024}^\dagger$ | – | – |
| GraphGPS (2022) | $0.7880_{\pm 0.0101}$ | $0.2907_{\pm 0.0028}$ | $0.8015_{\pm 0.0033}$ | $0.1894_{\pm 0.0024}$ |
| Specformer (2023) | $0.7889_{\pm 0.0124}$ | $\mathbf{0.2972}_{\pm 0.0023}$ | – | – |
| Graph ViT/MLP-Mixer (2023) | $0.7997_{\pm 0.0102}$ | – | – | – |
| Exphormer (2023) | $0.7834_{\pm 0.0044}$ | $0.2849_{\pm 0.0025}$ | – | – |
| GRIT (2023) | $0.7835_{\pm 0.0054}$ | $0.2362_{\pm 0.0020}$ | – | – |
| Subgraphormer (2024) | $\mathbf{0.8038}_{\pm 0.0192}$ | – | – | – |
| GECO (2024) | $0.7980_{\pm 0.0200}$ | $\mathbf{0.2961}_{\pm 0.0008}$ | $0.7982_{\pm 0.0042}$ | $\mathbf{0.1915}_{\pm 0.0020}$ |
| GSSC (2024b) | $\mathbf{0.8035}_{\pm 0.0142}$ | – | – | – |
| GCN | $0.7606_{\pm 0.0097}$ | $0.2020_{\pm 0.0024}$ | $0.6839_{\pm 0.0084}$ | $0.1507_{\pm 0.0018}$ |
| **GCN$^+$** | $0.8012_{\pm 0.0124}$ **5.4%↑** | $0.2721_{\pm 0.0046}$ **34.7%↑** | $\mathbf{0.8077}_{\pm 0.0041}$ **18.1%↑** | $0.1787_{\pm 0.0026}$ **18.6%↑** |
| GIN | $0.7835_{\pm 0.0125}$ | $0.2266_{\pm 0.0028}$ | $0.6892_{\pm 0.0100}$ | $0.1495_{\pm 0.0023}$ |
| **GIN$^+$** | $0.7928_{\pm 0.0099}$ **1.2%↑** | $0.2703_{\pm 0.0024}$ **19.3%↑** | $\mathbf{0.8107}_{\pm 0.0053}$ **17.7%↑** | $0.1803_{\pm 0.0019}$ **20.6%↑** |
| GatedGCN | $0.7687_{\pm 0.0136}$ | $0.2670_{\pm 0.0020}$ | $0.7531_{\pm 0.0083}$ | $0.1606_{\pm 0.0015}$ |
| **GatedGCN$^+$** | $\mathbf{0.8040}_{\pm 0.0164}$ **4.6%↑** | $\mathbf{0.2981}_{\pm 0.0024}$ **11.6%↑** | $\mathbf{0.8258}_{\pm 0.0055}$ **9.7%↑** | $\mathbf{0.1896}_{\pm 0.0024}$ **18.1%↑** |
| Time (epoch/s) of GraphGPS | 96s | 196s | 276s | 1919s |
| Time (epoch/s) of **GCN$^+$** | **16s** | **91s** | **178s** | **476s** |

*Table 5.* Ablation study on GNN Benchmark (Dwivedi et al., 2023) (%). - indicates that the corresponding hyperparameter is not used in GNN$^+$, as it empirically leads to inferior performance.

| Metric | ZINC MAE ↓ | MNIST Accuracy ↑ | CIFAR10 Accuracy ↑ | PATTERN Accuracy ↑ | CLUSTER Accuracy ↑ |
|---|---|---|---|---|---|
| **GCN$^+$** | $\mathbf{0.076}_{\pm 0.009}$ | $\mathbf{98.382}_{\pm 0.095}$ | $\mathbf{69.824}_{\pm 0.413}$ | $\mathbf{87.021}_{\pm 0.095}$ | $\mathbf{77.109}_{\pm 0.872}$ |
| (-) Edge. | $0.135_{\pm 0.004}$ | $98.153_{\pm 0.042}$ | $68.256_{\pm 0.357}$ | $86.854_{\pm 0.054}$ | – |
| (-) Norm | $0.107_{\pm 0.011}$ | $97.886_{\pm 0.066}$ | $60.765_{\pm 0.829}$ | $52.769_{\pm 0.874}$ | $16.563_{\pm 0.134}$ |
| (-) Dropout | – | $97.897_{\pm 0.071}$ | $65.693_{\pm 0.461}$ | $86.764_{\pm 0.045}$ | $74.926_{\pm 0.469}$ |
| (-) RC | $0.159_{\pm 0.016}$ | $95.929_{\pm 0.169}$ | $58.186_{\pm 0.295}$ | $86.059_{\pm 0.274}$ | $16.508_{\pm 0.615}$ |
| (-) FFN | $0.132_{\pm 0.021}$ | $97.174_{\pm 0.063}$ | $63.573_{\pm 0.346}$ | $86.746_{\pm 0.088}$ | $72.606_{\pm 1.243}$ |
| (-) PE | $0.127_{\pm 0.010}$ | – | – | $85.597_{\pm 0.241}$ | $75.568_{\pm 1.147}$ |
| **GIN$^+$** | $\mathbf{0.065}_{\pm 0.004}$ | $\mathbf{98.285}_{\pm 0.103}$ | $\mathbf{69.592}_{\pm 0.287}$ | $\mathbf{86.842}_{\pm 0.048}$ | $\mathbf{74.794}_{\pm 0.213}$ |
| (-) Edge. | $0.122_{\pm 0.009}$ | $97.655_{\pm 0.075}$ | $68.196_{\pm 0.107}$ | $86.714_{\pm 0.036}$ | $65.895_{\pm 3.425}$ |
| (-) Norm | $0.096_{\pm 0.006}$ | $97.695_{\pm 0.065}$ | $64.918_{\pm 0.059}$ | $86.815_{\pm 0.855}$ | $72.119_{\pm 0.359}$ |
| (-) Dropout | – | $98.214_{\pm 0.064}$ | $66.638_{\pm 0.873}$ | $86.836_{\pm 0.053}$ | $73.316_{\pm 0.355}$ |
| (-) RC | $0.137_{\pm 0.031}$ | $97.675_{\pm 0.175}$ | $64.910_{\pm 0.102}$ | $86.645_{\pm 0.125}$ | $16.800_{\pm 0.088}$ |
| (-) FFN | $0.104_{\pm 0.003}$ | $11.350_{\pm 0.008}$ | $60.582_{\pm 0.395}$ | $58.511_{\pm 0.016}$ | $62.175_{\pm 2.895}$ |
| (-) PE | $0.123_{\pm 0.014}$ | – | – | $86.592_{\pm 0.049}$ | $73.925_{\pm 0.165}$ |
| **GatedGCN$^+$** | $\mathbf{0.077}_{\pm 0.005}$ | $\mathbf{98.712}_{\pm 0.137}$ | $\mathbf{77.218}_{\pm 0.381}$ | $\mathbf{87.029}_{\pm 0.037}$ | $\mathbf{79.128}_{\pm 0.235}$ |
| (-) Edge. | $0.119_{\pm 0.001}$ | $98.085_{\pm 0.045}$ | $72.128_{\pm 0.275}$ | $86.879_{\pm 0.017}$ | $76.075_{\pm 0.845}$ |
| (-) Norm | $0.088_{\pm 0.003}$ | $98.275_{\pm 0.045}$ | $71.995_{\pm 0.445}$ | $86.942_{\pm 0.023}$ | $78.495_{\pm 0.155}$ |
| (-) Dropout | $0.089_{\pm 0.002}$ | $98.225_{\pm 0.095}$ | $70.383_{\pm 0.429}$ | $86.802_{\pm 0.034}$ | $77.597_{\pm 0.126}$ |
| (-) RC | $0.106_{\pm 0.002}$ | $98.442_{\pm 0.067}$ | $75.149_{\pm 0.155}$ | $86.845_{\pm 0.025}$ | $16.670_{\pm 0.307}$ |
| (-) FFN | $0.098_{\pm 0.005}$ | $98.438_{\pm 0.151}$ | $76.243_{\pm 0.131}$ | $86.935_{\pm 0.025}$ | $78.975_{\pm 0.145}$ |
| (-) PE | $0.174_{\pm 0.009}$ | – | – | $85.595_{\pm 0.065}$ | $77.515_{\pm 0.265}$ |

MalNet-Tiny. Furthermore, on PascalVOC-SP and COCO-SP, GatedGCN$^+$ significantly improves performance, securing the third-best model ranking overall. These results highlight the potential of classic GNNs in capturing long-range interactions in graph-level tasks.

**Open Graph Benchmark (OGB), Table 4.** Finally, we test our method on four OGB datasets. As shown in Table 4, GatedGCN$^+$ consistently ranks among the top three models and achieves top performance on three out of the four datasets. On ogbg-ppa, GatedGCN$^+$ shows an improvement of approximately 9%, ranking first on the OGB leaderboard. On ogbg-molhiv and ogbg-molpcba, GatedGCN$^+$ even matches the performance of Graphormer and EGT pre-trained on other datasets. Additionally, on ogbg-code2, GatedGCN$^+$ secures the third-highest performance, underscoring the potential of GNNs for large-scale OGB datasets.

### 5.2. Ablation Study

To examine the unique contributions of different technique used in GNN$^+$, we conduct a series of ablation analysis by selectively removing elements such as edge feature module (Edge.), normalization (Norm), dropout, residual connections (RC), FFN, PE from GCN$^+$, GIN$^+$, and GatedGCN$^+$. The effect of these ablations is assessed across GNN Benchmark (see Table 5), LRGB, and OGB (see Table 6) datasets.

> Our ablation study demonstrates that each module incorporated in GNN$^+$—including edge feature integration, normalization, dropout, residual connections, FFN, and PE—is **indispensable**; the removal of any single component results in a degradation of overall performance.

*Table 6.* Ablation study on LRGB and OGB datasets. - indicates that the corresponding hyperparameter is not used in GNN$^+$, as it empirically leads to inferior performance.

| Metric | Peptides-func Avg. Precision ↑ | Peptides-struct MAE ↓ | PascalVOC-SP F1 score ↑ | COCO-SP F1 score ↑ | MalNet-Tiny Accuracy ↑ | ogbg-molhiv AUROC ↑ | ogbg-molpcba Avg. Precision ↑ | ogbg-ppa Accuracy ↑ | ogbg-code2 F1 score ↑ |
|---|---|---|---|---|---|---|---|---|---|
| **GCN$^+$** | **0.7261** ±0.0067 | **0.2421** ±0.0016 | **0.3357** ±0.0087 | **0.2733** ±0.0041 | **0.9354** ±0.0045 | **0.8012** ±0.0124 | **0.2721** ±0.0046 | **0.8077** ±0.0041 | **0.1787** ±0.0026 |
| (-) Edge. | 0.7191 ±0.0036 | – | 0.2942 ±0.0043 | 0.2219 ±0.0060 | 0.9292 ±0.0034 | 0.7714 ±0.0204 | 0.2628 ±0.0019 | 0.2994 ±0.0062 | 0.1785 ±0.0033 |
| (-) Norm | 0.7107 ±0.0027 | 0.2509 ±0.0026 | 0.1802 ±0.0111 | 0.2332 ±0.0079 | 0.9236 ±0.0054 | 0.7753 ±0.0049 | 0.2528 ±0.0016 | 0.6705 ±0.0104 | 0.1679 ±0.0027 |
| (-) Dropout | 0.6748 ±0.0055 | 0.2549 ±0.0025 | 0.3072 ±0.0069 | 0.2601 ±0.0046 | – | 0.7431 ±0.0185 | 0.2405 ±0.0047 | 0.7893 ±0.0052 | 0.1641 ±0.0043 |
| (-) RC | – | – | 0.2734 ±0.0036 | 0.1948 ±0.0096 | 0.8916 ±0.0048 | – | – | 0.7520 ±0.0157 | 0.1785 ±0.0029 |
| (-) FFN | – | – | 0.2786 ±0.0068 | 0.2314 ±0.0073 | 0.9118 ±0.0078 | 0.7432 ±0.0052 | 0.2621 ±0.0019 | 0.7672 ±0.0071 | 0.1594 ±0.0020 |
| (-) PE | 0.7069 ±0.0093 | 0.2447 ±0.0015 | – | – | – | 0.7593 ±0.0051 | 0.2667 ±0.0034 | – | – |
| **GIN$^+$** | **0.7059** ±0.0089 | **0.2429** ±0.0019 | **0.3189** ±0.0105 | **0.2483** ±0.0046 | **0.9325** ±0.0040 | **0.7928** ±0.0099 | **0.2703** ±0.0024 | **0.8107** ±0.0053 | **0.1803** ±0.0019 |
| (-) Edge. | 0.7033 ±0.0015 | 0.2442 ±0.0028 | 0.2956 ±0.0047 | 0.2259 ±0.0053 | 0.9286 ±0.0049 | 0.7597 ±0.0103 | 0.2702 ±0.0021 | 0.2789 ±0.0031 | 0.1752 ±0.0020 |
| (-) Norm | 0.6934 ±0.0077 | 0.2444 ±0.0015 | 0.2707 ±0.0037 | 0.2244 ±0.0063 | 0.9322 ±0.0025 | 0.7874 ±0.0114 | 0.2556 ±0.0026 | 0.6484 ±0.0246 | 0.1722 ±0.0034 |
| (-) Dropout | 0.6384 ±0.0094 | 0.2531 ±0.0030 | 0.3153 ±0.0113 | – | – | – | 0.2545 ±0.0068 | 0.7673 ±0.0059 | 0.1730 ±0.0018 |
| (-) RC | 0.6975 ±0.0038 | 0.2527 ±0.0015 | 0.2350 ±0.0044 | 0.1741 ±0.0085 | 0.9150 ±0.0047 | 0.7733 ±0.0122 | 0.1454 ±0.0061 | – | 0.1617 ±0.0026 |
| (-) FFN | – | – | 0.2393 ±0.0049 | 0.1599 ±0.0081 | 0.8944 ±0.0074 | – | 0.2534 ±0.0033 | 0.6676 ±0.0039 | 0.1491 ±0.0016 |
| (-) PE | 0.6855 ±0.0027 | 0.2455 ±0.0019 | 0.3141 ±0.0031 | – | – | 0.7791 ±0.0268 | 0.2601 ±0.0023 | – | – |
| **GatedGCN$^+$** | **0.7006** ±0.0033 | **0.2431** ±0.0020 | **0.4263** ±0.0057 | **0.3802** ±0.0015 | **0.9460** ±0.0057 | **0.8040** ±0.0164 | **0.2981** ±0.0024 | **0.8258** ±0.0055 | **0.1896** ±0.0024 |
| (-) Edge. | 0.6882 ±0.0028 | 0.2466 ±0.0018 | 0.3764 ±0.0117 | 0.3172 ±0.0109 | 0.9372 ±0.0062 | 0.7831 ±0.0157 | 0.2951 ±0.0028 | 0.0948 ±0.0000 | 0.1891 ±0.0021 |
| (-) Norm | 0.6733 ±0.0026 | 0.2474 ±0.0015 | 0.3628 ±0.0043 | 0.3527 ±0.0051 | 0.9326 ±0.0056 | 0.7879 ±0.0178 | 0.2748 ±0.0012 | 0.6864 ±0.0165 | 0.1743 ±0.0026 |
| (-) Dropout | 0.6695 ±0.0101 | 0.2508 ±0.0014 | 0.3389 ±0.0066 | 0.3393 ±0.0051 | – | – | 0.2582 ±0.0036 | 0.8088 ±0.0062 | 0.1724 ±0.0027 |
| (-) RC | – | 0.2498 ±0.0034 | 0.4075 ±0.0052 | 0.3475 ±0.0064 | 0.9402 ±0.0054 | 0.7833 ±0.0177 | 0.2897 ±0.0016 | 0.8099 ±0.0053 | 0.1844 ±0.0025 |
| (-) FFN | – | – | – | 0.3508 ±0.0049 | 0.9364 ±0.0059 | – | 0.2875 ±0.0022 | – | 0.1718 ±0.0024 |
| (-) PE | 0.6729 ±0.0084 | 0.2461 ±0.0025 | 0.4052 ±0.0031 | – | – | 0.7771 ±0.0057 | 0.2813 ±0.0022 | – | – |

**Observation 1: The integration of edge features is particularly effective in molecular and image superpixel datasets, where these features carry critical information.**

In molecular graphs such as ZINC and ogbg-molhiv, edge features represent chemical bond information, which is essential for molecular properties. Removing this module leads to a significant performance drop. In protein networks ogbg-ppa, edges represent normalized associations between proteins. Removing the edge feature module results in a substantial accuracy decline, ranging from 0.5083 to 0.7310 for classic GNNs. Similarly, in image superpixel datasets like CIFAR-10, PascalVOC-SP, and COCO-SP, edge features encode spatial relationships between superpixels, which are crucial for maintaining image coherence. However, in code graphs such as ogbg-code2 and MalNet-Tiny, where edges represent call types, edge features are less relevant to the prediction tasks, and their removal has minimal impact.

**Observation 2: Normalization tends to have a greater impact on larger-scale datasets, whereas its impact is less significant on smaller datasets.**

For large-scale datasets such as CIFAR 10, COCO-SP, and the OGB datasets, removing normalization leads to significant performance drops. Specifically, on ogbg-ppa, which has 158,100 graphs, ablating normalization results in an accuracy drop of around 15% for three classic GNNs. This result is consistent with Luo et al. (2024b), who found that normalization is more important for GNNs in node classification on large graphs. In such datasets, where node feature distributions are more complex, normalizing node

embeddings is essential for stabilizing the training process.

**Observation 3: Dropout proves advantageous for most datasets, with a very low dropout rate being sufficient and optimal**.

Our analysis highlights the crucial role of dropout in maintaining the performance of classic GNNs on GNN Benchmark and LRGB and large-scale OGB datasets, with its ablation causing significant declines—for instance, an 8.8% relative decrease for GatedGCN$^+$ on CIFAR-10 and a 20.4% relative decrease on PascalVOC-SP. This trend continues in large-scale OGB datasets, where removing dropout results in a 5–13% performance drop across 3 classic GNNs on ogbg-molpcba. Notably, 97% of the optimal dropout rates are $\leq 0.2$, and 64% are $\leq 0.1$, indicating that a very low dropout rate is both sufficient and optimal for graph-level tasks. Interestingly, this finding for graph-level tasks contrasts with Luo et al. (2024b)'s observations for node-level tasks, where a higher dropout rate is typically required.

**Observation 4: Residual connections are generally essential, except in shallow GNNs applied to small graphs.**

Removing residual connections generally leads to significant performance drops across datasets, with the only exceptions being found in the peptide datasets. Although similar in the number of nodes to CLUSTER and PATTERN, peptide datasets involve GNNs with only 3-5 layers, while the others use deeper networks with over 10 layers. For shallow networks in small graphs, residual connections may not be as beneficial and can even hurt performance by disrupting feature flow. In contrast, deeper networks in larger graphs

rely on residual connections to maintain gradient flow and enable stable, reliable long-range information exchange.

### Observation 5: FFN is crucial for GIN$^+$ and GCN$^+$, greatly impacting their performance across datasets.

Ablating FFN leads to substantial performance declines for GIN$^+$ and GCN$^+$ across almost all datasets, highlighting its essential role in graph-level tasks. Notably, on MNIST, removing FFN leads to an 88% relative accuracy drop for GIN$^+$. This is likely because the architectures of GIN$^+$ and GCN$^+$ rely heavily on FFN for learning complex node feature representations. In contrast, GatedGCN$^+$ uses gating mechanisms to adaptively adjust the importance of neighboring nodes' information, reducing the need for additional feature transformations. The only exceptions are observed in the peptides datasets, where FFN is not used in all three models. This may be due to the shallow GNN architecture, where complex feature transformations are less necessary.

### Observation 6: PE is particularly effective for small-scale datasets, but negligible for large-scale datasets.

Removing PE significantly reduces performance for classic GNNs on small-scale datasets like ZINC, PATTERN, CLUS-TER, Peptides-func, and ogbg-molhiv, which only contain 10,000-40,000 graphs. By contrast, on large-scale datasets like ogbg-code2, ogbg-molpcba, ogbg-ppa, and COCO-SP (over 100,000 graphs), the impact of PE is less pronounced. This may be because smaller datasets rely more on PE to capture graph structure, whereas larger datasets benefit from the abundance of data, reducing the need for PE.

## 6. Related Work

Our work relates closely to recent efforts in benchmarking GNNs. As node-level benchmarks have received significantly more attention, several studies such as (Shchur et al., 2018; Wang et al., 2021; Lv et al., 2021; Platonov et al., 2023) have examined the reliability of existing evaluation protocols, revealing that many newly proposed architectures fail to outperform simple baselines under fair conditions. Building on this line of inquiry, Luo et al. (2024b) rigorously showed that, with the same hyperparameter search space, classic GNNs can match or even surpass SOTA models across 18 widely used node classification datasets.

In contrast, our study targets graph-level tasks. Errica et al. (2020) highlights the importance of rigorous evaluation protocols and shows that many claimed improvements vanish under fair settings. Hu et al. (2020) introduces the OGB benchmark to standardize large-scale graph evaluations. Dwivedi et al. (2023) introduces a diverse suite of graphs for fair architectural comparisons under fixed parameter budgets, and Dwivedi et al. (2022) proposes the LRGB to assess models on long-range dependency tasks. Tönshoff

et al. (2023) further investigates LRGB and shows that classic GNNs can close the gap with GTs under proper tuning. Grötschla et al. (2024) offers a systematic study of positional encoding. However, these studies have been constrained in both scope and comprehensiveness, primarily due to the limited number and diversity of datasets used, as well as an incomplete examination of GNN hyperparameters.

## 7. Limitations and Future Work

While our study provides strong empirical evidence that enhanced classic GNNs can match or even surpass GTs in graph-level tasks on existing benchmarks, it is important to acknowledge that these findings are derived solely from empirical evaluations and lack a theoretical explanation for the observed advantage. Moreover, as highlighted in recent discussions (Bechler-Speicher et al., 2025), current graph datasets may not adequately capture the complexity and diversity inherent in real-world graph problems. Consequently, although our results demonstrate the competitiveness of classic GNNs under current datasets and evaluation settings, their relative advantage may shift as more challenging and application-driven benchmarks are developed.

A deeper theoretical understanding is needed to elucidate the role of each architectural component within GNN$^+$ and their interactions, particularly regarding the model's expressiveness, representational behavior, and its capacity to address oversmoothing and oversquashing issues. These directions represent promising avenues for future research.

## 8. Conclusion

This study highlights the often-overlooked potential of classic GNNs in addressing graph-level tasks. By integrating six widely used techniques into a unified GNN$^+$ framework, we enhance three classic GNNs (GCN, GIN, and GatedGCN) for graph-level tasks. Comprehensive evaluations on 14 benchmark datasets reveal that these enhanced GNNs can match or even outperform SOTA GTs, while also demonstrating greater computational efficiency. These findings challenge the prevailing belief that GTs are inherently superior, reaffirming the effectiveness of simple GNN structures as powerful models for graph-level tasks.

### Acknowledgments

We sincerely thank the anonymous area chair for their thoughtful evaluation and strong support, which gave us the opportunity to share this work with the community. We are also grateful to all anonymous reviewers for their constructive and insightful feedback. We sincerely thank Yiwen Sun for her invaluable help with refining the writing and Lu Fan for her help with creating Figure 1.

This work received support from National Key R&D Program of China (2021YFB3500700), NSFC Grant 62172026, National Social Science Fund of China 22&ZD153, the Fundamental Research Funds for the Central Universities, State Key Laboratory of Complex & Critical Software Environment (SKLCCSE), and the HK PolyU Grant P0051029. Lei Shi is with School of Computer Science and Engineering, Beihang University, and the State Key Laboratory of Complex & Critical Software Environment.

## Impact Statement

This paper presents an empirical benchmarking study that re-evaluates the performance of classic GNNs on graph-level tasks. By systematically integrating widely used techniques into a unified framework (GNN$^+$), the study demonstrates that classic GNNs can achieve performance comparable to SOTA models. The findings challenge common assumptions in the field and provide practical guidance for future model selection and design. As this is a benchmarking effort, we do not foresee any immediate societal consequences that must be specifically highlighted.

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

# A. Datasets and Experimental Details

## A.1. Computing Environment

Our implementation is based on PyG (Fey & Lenssen, 2019). The experiments are conducted on a single workstation with 8 RTX 3090 GPUs.

## A.2. Datasets

Table 7 presents a summary of the statistics and characteristics of the datasets.

- **GNN Benchmark** (Dwivedi et al., 2023). **ZINC** contains molecular graphs with node features representing atoms and edge features representing bonds The task is to regress the constrained solubility (logP) of the molecule. **MNIST** and **CIFAR10** are adapted from image classification datasets, where each image is represented as an 8-nearest-neighbor graph of SLIC superpixels, with nodes representing superpixels and edges representing spatial relationships. The 10-class classification tasks follow the original image classification tasks. **PATTERN** and **CLUSTER** are synthetic datasets sampled from the Stochastic Block Model (SBM) for inductive node classification, with tasks involving sub-graph pattern recognition and cluster ID inference. For all datasets, we adhere to the respective training protocols and standard evaluation splits (Dwivedi et al., 2023).
- **Long-Range Graph Benchmark (LRGB)** (Dwivedi et al., 2022; Freitas & Dong, 2021). **Peptides-func** and **Peptides-struct** are atomic graphs of peptides from SATPdb, with tasks of multi-label graph classification into 10 peptide functional classes and graph regression for 11 3D structural properties, respectively. **PascalVOC-SP** and **COCO-SP** are node classification datasets derived from the Pascal VOC and MS COCO images by SLIC superpixelization, where each superpixel node belongs to a particular object class. We did not use PCQM-Contact in (Dwivedi et al., 2022) as its download link was no longer valid. **MalNet-Tiny** (Freitas & Dong, 2021) is a subset of MalNet with 5,000 function call graphs (FCGs) from Android APKs, where the task is to predict software type based on structure alone. For each dataset, we follow standard training protocols and splits (Dwivedi et al., 2022; Freitas & Dong, 2021).
- **Open Graph Benchmark (OGB)** (Hu et al., 2020). We also consider a collection of larger-scale datasets from OGB, containing graphs in the range of hundreds of thousands to millions: **ogbg-molhiv** and **ogbg-molpcba** are molecular property prediction datasets from MoleculeNet. ogbg-molhiv involves binary classification of HIV inhibition, while ogbg-molpcba predicts results of 128 bioassays in a multi-task setting. **ogbg-ppa** contains protein-protein association networks, where nodes represent proteins and edges encode normalized associations between them; the task is to classify the origin of the network among 37 taxonomic groups. **ogbg-code2** consists of abstract syntax trees (ASTs) from Python source code, with the task of predicting the first 5 subtokens of the function's name. We maintain all the OGB standard evaluation settings (Hu et al., 2020).

*Table 7.* Overview of the datasets used for graph-level tasks (Dwivedi et al., 2023; 2022; Hu et al., 2020; Freitas & Dong, 2021).

| Dataset | # graphs | Avg. # nodes | Avg. # edges | # node/edge feats | Prediction level | Prediction task | Metric |
|---|---|---|---|---|---|---|---|
| ZINC | 12,000 | 23.2 | 24.9 | 28/1 | graph | regression | MAE |
| MNIST | 70,000 | 70.6 | 564.5 | 3/1 | graph | 10-class classif. | Accuracy |
| CIFAR10 | 60,000 | 117.6 | 941.1 | 5/1 | graph | 10-class classif. | Accuracy |
| PATTERN | 14,000 | 118.9 | 3,039.3 | 3/1 | inductive node | binary classif. | Accuracy |
| CLUSTER | 12,000 | 117.2 | 2,150.9 | 7/1 | inductive node | 6-class classif. | Accuracy |
| Peptides-func | 15,535 | 150.9 | 307.3 | 9/3 | graph | 10-task classif. | Avg. Precision |
| Peptides-struct | 15,535 | 150.9 | 307.3 | 9/3 | graph | 11-task regression | MAE |
| PascalVOC-SP | 11,355 | 479.4 | 2,710.5 | 14/2 | inductive node | 21-class classif. | F1 score |
| COCO-SP | 123,286 | 476.9 | 2,693.7 | 14/2 | inductive node | 81-class classif. | F1 score |
| MalNet-Tiny | 5,000 | 1,410.3 | 2,859.9 | 5/1 | graph | 5-class classif. | Accuracy |
| ogbg-molhiv | 41,127 | 25.5 | 27.5 | 9/3 | graph | binary classif. | AUROC |
| ogbg-molpcba | 437,929 | 26.0 | 28.1 | 9/3 | graph | 128-task classif. | Avg. Precision |
| ogbg-ppa | 158,100 | 243.4 | 2,266.1 | 1/7 | graph | 37-task classif. | Accuracy |
| ogbg-code2 | 452,741 | 125.2 | 124.2 | 2/2 | graph | 5 token sequence | F1 score |

## A.3. Hyperparameters and Reproducibility

Please note that we mainly follow the experiment settings of GraphGPS (Rampášek et al., 2022; Tönshoff et al., 2023). For the hyperparameter selections of classic GNNs, in addition to what we have covered, we list other settings in Tables 8, 9, 10,

 Further details regarding hyperparameters can be found in our code.

In all experiments, we use the validation set to select the best hyperparameters. **GNN**$^+$ denotes enhanced implementation of the GNN model.

Our code is available under the MIT License.

*Table 8.* Hyperparameter settings of GCN$^+$ on benchmarks from (Dwivedi et al., 2023).

| Hyperparameter | ZINC | MNIST | CIFAR10 | PATTERN | CLUSTER |
|---|---|---|---|---|---|
| # GNN Layers | 12 | 6 | 5 | 12 | 12 |
| Edge Feature Module | True | True | True | True | False |
| Normalization | BN | BN | BN | BN | BN |
| Dropout | 0.0 | 0.15 | 0.05 | 0.05 | 0.1 |
| Residual Connections | True | True | True | True | True |
| FFN | True | True | True | True | True |
| PE | RWSE-32 | False | False | RWSE-32 | RWSE-20 |
| Hidden Dim | 64 | 60 | 65 | 90 | 90 |
| Graph Pooling | add | mean | mean | – | – |
| Batch Size | 32 | 16 | 16 | 32 | 16 |
| Learning Rate | 0.001 | 0.0005 | 0.001 | 0.001 | 0.001 |
| # Epochs | 2000 | 200 | 200 | 200 | 100 |
| # Warmup Epochs | 50 | 5 | 5 | 5 | 5 |
| Weight Decay | 1e-5 | 1e-5 | 1e-5 | 1e-5 | 1e-5 |
| # Parameters | 260,177 | 112,570 | 114,345 | 517,219 | 516,674 |
| Time (epoch) | 7.6s | 60.1s | 40.2s | 19.5s | 29.7s |

*Table 9.* Hyperparameter settings of GCN$^+$ on LRGB and OGB datasets.

| Hyperparameter | Peptides-func | Peptides-struct | PascalVOC-SP | COCO-SP | MalNet-Tiny | ogbg-molhiv | ogbg-molpcba | ogbg-ppa | ogbg-code2 |
|---|---|---|---|---|---|---|---|---|---|
| # GNN Layers | 3 | 5 | 14 | 18 | 8 | 4 | 10 | 4 | 4 |
| Edge Feature Module | True | False | True | True | True | True | True | True | True |
| Normalization | BN | BN | BN | BN | BN | BN | BN | BN | BN |
| Dropout | 0.2 | 0.2 | 0.1 | 0.05 | 0.0 | 0.1 | 0.2 | 0.2 | 0.2 |
| Residual Connections | False | False | True | True | True | False | False | True | True |
| FFN | False | False | True | True | True | True | True | True | True |
| PE | RWSE-32 | RWSE-32 | False | False | False | RWSE-20 | RWSE-16 | False | False |
| Hidden Dim | 275 | 255 | 85 | 70 | 110 | 256 | 512 | 512 | 512 |
| Graph Pooling | mean | mean | – | – | max | mean | mean | mean | mean |
| Batch Size | 16 | 32 | 50 | 50 | 16 | 32 | 512 | 32 | 32 |
| Learning Rate | 0.001 | 0.001 | 0.001 | 0.001 | 0.0005 | 0.0001 | 0.0005 | 0.0003 | 0.0001 |
| # Epochs | 300 | 300 | 200 | 300 | 150 | 100 | 100 | 400 | 30 |
| # Warmup Epochs | 5 | 5 | 10 | 10 | 10 | 5 | 5 | 10 | 2 |
| Weight Decay | 0.0 | 0.0 | 0.0 | 0.0 | 1e-5 | 1e-5 | 1e-5 | 1e-5 | 1e-6 |
| # Parameters | 507,351 | 506,127 | 520,986 | 460,611 | 494,235 | 1,407,641 | 13,316,700 | 5,549,605 | 23,291,826 |
| Time (epoch) | 6.9s | 6.6s | 12.5s | 162.5s | 6.6s | 16.3s | 91.4s | 178.2s | 476.3s |

*Table 10.* Hyperparameter settings of GIN$^+$ on benchmarks from (Dwivedi et al., 2023).

| Hyperparameter | ZINC | MNIST | CIFAR10 | PATTERN | CLUSTER |
|---|---|---|---|---|---|
| # GNN Layers | 12 | 5 | 5 | 8 | 10 |
| Edge Feature Module | True | True | True | True | True |
| Normalization | BN | BN | BN | BN | BN |
| Dropout | 0.0 | 0.1 | 0.05 | 0.05 | 0.05 |
| Residual Connections | True | True | True | True | True |
| FFN | True | True | True | True | True |
| PE | RWSE-20 | False | False | RWSE-32 | RWSE-20 |
| Hidden Dim | 80 | 60 | 60 | 100 | 90 |
| Graph Pooling | sum | mean | mean | – | – |
| Batch Size | 32 | 16 | 16 | 32 | 16 |
| Learning Rate | 0.001 | 0.001 | 0.001 | 0.001 | 0.0005 |
| # Epochs | 2000 | 200 | 200 | 200 | 100 |
| # Warmup Epochs | 50 | 5 | 5 | 5 | 5 |
| Weight Decay | 1e-5 | 1e-5 | 1e-5 | 1e-5 | 1e-5 |
| # Parameters | 477,241 | 118,990 | 115,450 | 511,829 | 497,594 |
| Time (epoch) | 9.4s | 56.8s | 46.3s | 18.5s | 20.5s |

*Table 11.* Hyperparameter settings of GIN$^+$ on LRGB and OGB datasets.

| Hyperparameter | Peptides-func | Peptides-struct | PascalVOC-SP | COCO-SP | MalNet-Tiny | ogbg-molhiv | ogbg-molpcba | ogbg-ppa | ogbg-code2 |
|---|---|---|---|---|---|---|---|---|---|
| # GNN Layers | 3 | 5 | 16 | 16 | 5 | 3 | 16 | 5 | 4 |
| Edge Feature Module | True | True | True | True | True | True | True | True | True |
| Normalization | BN | BN | BN | BN | BN | BN | BN | BN | BN |
| Dropout | 0.2 | 0.2 | 0.1 | 0.0 | 0.0 | 0.0 | 0.3 | 0.15 | 0.1 |
| Residual Connections | True | True | True | True | True | True | True | False | True |
| FFN | False | False | True | True | True | False | True | True | True |
| PE | RWSE-32 | RWSE-32 | RWSE-32 | False | False | RWSE-20 | RWSE-16 | False | False |
| Hidden Dim | 240 | 200 | 70 | 70 | 130 | 256 | 300 | 512 | 512 |
| Graph Pooling | mean | mean | – | – | max | mean | mean | mean | mean |
| Batch Size | 16 | 32 | 50 | 50 | 16 | 32 | 512 | 32 | 32 |
| Learning Rate | 0.0005 | 0.001 | 0.001 | 0.001 | 0.0005 | 0.0001 | 0.0005 | 0.0003 | 0.0001 |
| # Epochs | 300 | 250 | 200 | 300 | 150 | 100 | 100 | 300 | 30 |
| # Warmup Epochs | 5 | 5 | 10 | 10 | 10 | 5 | 5 | 10 | 2 |
| Weight Decay | 0.0 | 0.0 | 0.0 | 0.0 | 1e-5 | 1e-5 | 1e-5 | 1e-5 | 1e-6 |
| # Parameters | 506,126 | 518,127 | 486,039 | 487,491 | 514,545 | 481,433 | 8,774,720 | 8,173,605 | 24,338,354 |
| Time (epoch) | 7.4s | 6.1s | 14.8s | 169.2s | 5.9s | 10.9s | 89.2s | 213.9s | 489.8s |

*Table 12.* Hyperparameter settings of GatedGCN$^+$ on benchmarks from (Dwivedi et al., 2023).

| Hyperparameter | ZINC | MNIST | CIFAR10 | PATTERN | CLUSTER |
|---|---|---|---|---|---|
| # GNN Layers | 9 | 10 | 10 | 12 | 16 |
| Edge Feature Module | True | True | True | True | True |
| Normalization | BN | BN | BN | BN | BN |
| Dropout | 0.05 | 0.05 | 0.15 | 0.2 | 0.2 |
| Residual Connections | True | True | True | True | True |
| FFN | True | True | True | True | True |
| PE | RWSE-20 | False | False | RWSE-32 | RWSE-20 |
| Hidden Dim | 70 | 35 | 35 | 64 | 56 |
| Graph Pooling | sum | mean | mean | – | – |
| Batch Size | 32 | 16 | 16 | 32 | 16 |
| Learning Rate | 0.001 | 0.001 | 0.001 | 0.0005 | 0.0005 |
| # Epochs | 2000 | 200 | 200 | 200 | 100 |
| # Warmup Epochs | 50 | 5 | 5 | 5 | 5 |
| Weight Decay | 1e-5 | 1e-5 | 1e-5 | 1e-5 | 1e-5 |
| # Parameters | 413,355 | 118,940 | 116,490 | 466,001 | 474,574 |
| Time (epoch) | 10.5s | 137.9s | 115.0s | 32.6s | 34.1s |

*Table 13.* Hyperparameter settings of GatedGCN$^+$ on LRGB and OGB datasets.

| Hyperparameter | Peptides-func | Peptides-struct | PascalVOC-SP | COCO-SP | MalNet-Tiny | ogbg-molhiv | ogbg-molpcba | ogbg-ppa | ogbg-code2 |
|---|---|---|---|---|---|---|---|---|---|
| # GNN Layers | 5 | 4 | 12 | 20 | 6 | 3 | 10 | 4 | 5 |
| Edge Feature Module | True | True | True | True | True | True | True | True | True |
| Normalization | BN | BN | BN | BN | BN | BN | BN | BN | BN |
| Dropout | 0.05 | 0.2 | 0.15 | 0.05 | 0.0 | 0.0 | 0.2 | 0.15 | 0.2 |
| Residual Connections | False | True | True | True | True | True | True | True | True |
| FFN | False | False | False | True | True | False | True | False | True |
| PE | RWSE-32 | RWSE-32 | RWSE-32 | False | False | RWSE-20 | RWSE-16 | False | False |
| Hidden Dim | 135 | 145 | 95 | 52 | 100 | 256 | 256 | 512 | 512 |
| Graph Pooling | mean | mean | – | – | max | mean | mean | mean | mean |
| Batch Size | 16 | 32 | 32 | 50 | 16 | 32 | 512 | 32 | 32 |
| Learning Rate | 0.0005 | 0.001 | 0.001 | 0.001 | 0.0005 | 0.0001 | 0.0005 | 0.0003 | 0.0001 |
| # Epochs | 300 | 300 | 200 | 300 | 150 | 100 | 100 | 300 | 30 |
| # Warmup Epochs | 5 | 5 | 10 | 10 | 10 | 5 | 5 | 10 | 2 |
| Weight Decay | 0.0 | 0.0 | 0.0 | 0.0 | 1e-5 | 1e-5 | 1e-5 | 1e-5 | 1e-6 |
| # Parameters | 521,141 | 492,897 | 559,094 | 508,589 | 550,905 | 1,076,633 | 6,016,860 | 5,547,557 | 29,865,906 |
| Time (epoch) | 17.3s | 8.0s | 21.3s | 208.8s | 8.9s | 15.1s | 85.1s | 479.8s | 640.1s |

# B. Additional Experimental Results

## B.1. Hyperparameter Sensitivity Analysis

To further understand the robustness of GNN$^+$, we conduct a comprehensive sensitivity analysis focusing on two key hyperparameters: the number of layers and the dropout rate.

**Layer Depth.** We investigate the impact of the number of network layers on three datasets: PATTERN and CLUSTER, and PascalVOC-SP. As shown in Figure 3, GNN$^+$ achieves strong and stable performance across a wide depth range. On PATTERN, both GCN$^+$ and GatedGCN$^+$ reach peak performance at 12 layers. In contrast, on CLUSTER and PascalVOC-SP, performance continues to improve as the number of layers increases, suggesting that GNN$^+$ is capable of leveraging deeper architectures when appropriate.

Importantly, when residual connections are disabled (bottom row), we observe significant performance degradation as the number of layers increases, especially beyond 8 layers. This is most pronounced on CLUSTER and PascalVOC-SP, where over-smoothing and over-squashing effects become severe in the absence of residual pathways. These results confirm that residual connections play a crucial role in stabilizing deep message propagation in GNN$^+$.

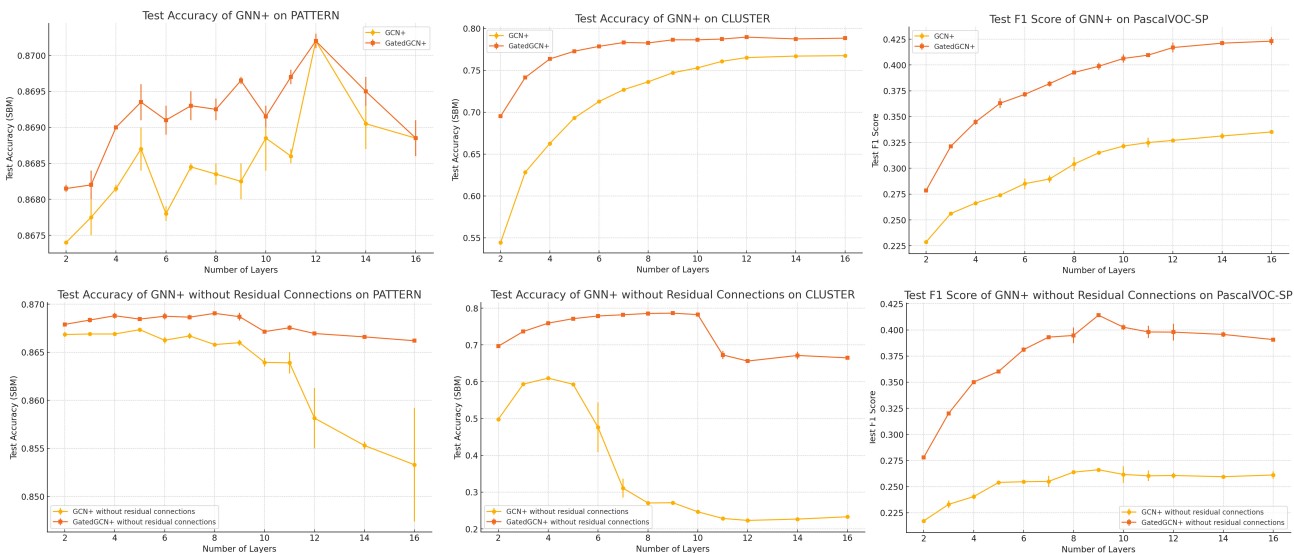

*Figure 3.* Experiments on the impact of the number of layers in GNN$^+$ on PATTERN, CLUSTER, and PascalVOC-SP. Top row: performance with residual connections; bottom row: performance without residual connections.

**Dropout Rate.** We conduct additional experiments to analyze the effect of dropout rates (ranging from 0.0 to 0.5) on four datasets: CLUSTER, PascalVOC-SP, Peptides-struct, and Peptides-func. As shown in Figure 4, the optimal dropout rate lies in the range of 0.1 to 0.2. Performance degrades significantly when the dropout rate reaches 0.3 or higher across all datasets. For instance, on Peptides-struct, the MAE increases sharply beyond 0.2 dropout, and on CLUSTER, accuracy collapses at 0.5. This suggests that excessive dropout disrupts the message-passing process in graph-level tasks.

**Binary Hyperparameters.** For components that can be enabled or disabled, such as edge features, normalization, residual connections, FFN, and positional encoding, we refer readers to the ablation results in Tables 5 and 6.

## B.2. Performance of a Fixed GNN$^+$ Configuration

We conduct an additional experiment using a fixed architecture of GNN$^+$, in which all six techniques are integrated. This model referred to as GNN$^+$ (fixed) is applied uniformly across datasets without dataset-specific tuning. The results are summarized in Table 14. The performance of GNN$^+$ (fixed) is comparable to the best configurations reported in the main paper, further demonstrating the model's practicality and robustness without requiring extensive tuning.

*Table 14.* Performance of GNN$^+$ with fixed configuration (i.e., all six techniques integrated) across 9 datasets.

| | PascalVOC-SP | COCO-SP | MalNet-Tiny | ogbg-molpcba | ogbg-code2 | MNIST | CIFAR10 | PATTERN | CLUSTER |
|---|---|---|---|---|---|---|---|---|---|
| GCN$^+$ | $0.3357_{\pm 0.0087}$ | $0.2733_{\pm 0.0041}$ | $0.9354_{\pm 0.0045}$ | $0.2721_{\pm 0.0046}$ | $0.1787_{\pm 0.0026}$ | $98.382_{\pm 0.095}$ | $69.824_{\pm 0.413}$ | $87.021_{\pm 0.095}$ | $77.109_{\pm 0.872}$ |
| GCN$^+$ (fixed) | $0.3341_{\pm 0.0055}$ | $0.2716_{\pm 0.0034}$ | $0.9235_{\pm 0.0060}$ | $0.2694_{\pm 0.0059}$ | $0.1784_{\pm 0.0029}$ | $98.257_{\pm 0.063}$ | $69.436_{\pm 0.265}$ | $87.021_{\pm 0.095}$ | $76.352_{\pm 0.757}$ |
| GatedGCN$^+$ | $0.4263_{\pm 0.0057}$ | $0.3802_{\pm 0.0015}$ | $0.9460_{\pm 0.0057}$ | $0.2981_{\pm 0.0024}$ | $0.1896_{\pm 0.0024}$ | $98.712_{\pm 0.137}$ | $77.218_{\pm 0.381}$ | $87.029_{\pm 0.037}$ | $79.128_{\pm 0.235}$ |
| GatedGCN$^+$ (fixed) | $0.4204_{\pm 0.0061}$ | $0.3774_{\pm 0.0028}$ | $0.9450_{\pm 0.0045}$ | $0.2981_{\pm 0.0024}$ | $0.1889_{\pm 0.0018}$ | $98.712_{\pm 0.137}$ | $77.218_{\pm 0.381}$ | $87.029_{\pm 0.037}$ | $79.128_{\pm 0.235}$ |

## B.3. Additional Baselines

**Comparison with Fast Linear Models.** To assess the performance-efficiency trade-off, we compare GCN$^+$ with two recent linear-time graph classification baselines, LDP (Cai & Wang, 2018) and HRN (Wu et al., 2022), on both small-scale and large-scale datasets, using the same training protocols and hyperparameter search spaces.

We first evaluate all models on five small-scale graph classification datasets used in (Cai & Wang, 2018; Wu et al., 2022): IMDB-B, IMDB-M, COLLAB, MUTAG, and PTC. These datasets contain between 300 and 5,000 graphs. As shown in Table 15, GCN$^+$ achieves accuracy comparable to or exceeding that of LDP and HRN on all datasets. Given the small scale,

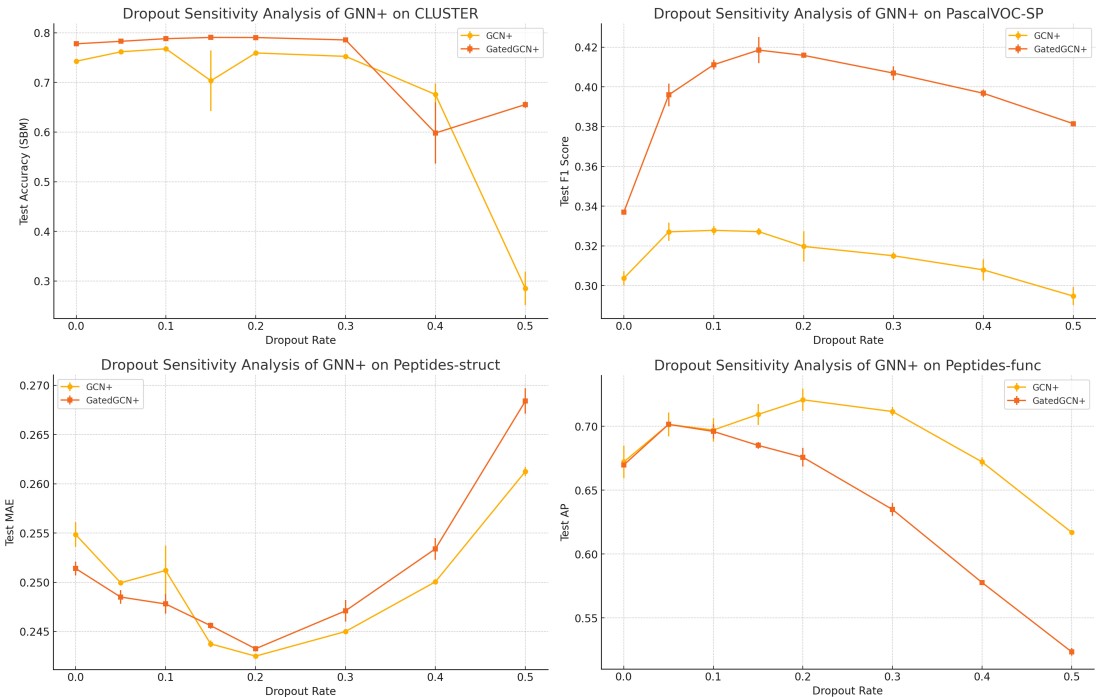

*Figure 4.* Sensitivity analysis of dropout rates in GNN$^+$ on CLUSTER, PascalVOC-SP, Peptides-struct, and Peptides-func.

training times are negligible across models.

*Table 15.* Accuracy (%) on small-scale graph classification datasets.

|  | IMDB-B | IMDB-M | COLLAB | MUTAG | PTC |
|---|---|---|---|---|---|
| # graphs | 1000 | 1500 | 5000 | 188 | 344 |
| LDP | 75.4 | 50.0 | 78.1 | 90.3 | 64.5 |
| HRN | 77.5 | 52.8 | 81.8 | 90.4 | 65.7 |
| GCN$^+$ | 76.9 | 52.3 | 80.9 | 90.1 | 66.6 |

To examine scalability, we evaluate all models on two OGB datasets: ogbg-molhiv and ogbg-molpcba. While LDP and HRN exhibit lower training costs, their predictive performance is significantly lower than GCN$^+$. Table 16 summarizes this trade-off, demonstrating the importance of benchmarking on large datasets for assessing practical utility.

*Table 16.* Performance and training cost on OGB molecular property prediction datasets.

|  | ogbg-molhiv | | ogbg-molpcba | |
|---|---|---|---|---|
|  | AUROC ↑ | Time (s/epoch) ↓ | Avg. Precision ↑ | Time (s/epoch) ↓ |
| LDP | $0.7121_{\pm 0.0105}$ | 5 | $0.1243_{\pm 0.0031}$ | 31 |
| HRN | $0.7587_{\pm 0.0147}$ | 9 | $0.2274_{\pm 0.0043}$ | 68 |
| GCN$^+$ | $0.8012_{\pm 0.0124}$ | 16 | $0.2721_{\pm 0.0046}$ | 91 |

**Comparison with SOTA GNNs for Over-smoothing and Over-squashing.** We further compare GCN$^+$ with two recently proposed models designed to mitigate over-smoothing and over-squashing: Multi-Track GCN (MTGCN) (Pei et al., 2024) and Cooperative GNN (CO-GNN) (Finkelshtein et al., 2024). All models are evaluated on Peptides-func and Peptides-struct using consistent training protocols and hyperparameter search spaces.

For CO-GNN, we tune the message-passing scheme (SUMGNN, MEANGNN, GCN, GIN, GAT) as recommended by the original paper. For MTGCN, we optimize the number of message-passing stages (from 1 to 4). Results are shown in

Table 17. GCN$^+$ consistently outperforms both models on both datasets, confirming its effectiveness in addressing these limitations while retaining architectural simplicity.

*Table 17.* Comparison with models addressing over-smoothing and over-squashing.

|  | Peptides-func | Peptides-struct |
|---|---|---|
| MTGCN | $0.6936 \pm 0.0089$ | $0.2461 \pm 0.0019$ |
| CO-GNN | $0.7012 \pm 0.0106$ | $0.2503 \pm 0.0025$ |
| GCN$^+$ | $0.7261 \pm 0.0067$ | $0.2421 \pm 0.0016$ |

## B.4. Visualization

To gain further insights into the representational quality of GNN$^+$, we visualize the learned graph embeddings using t-SNE. Specifically, we compare embeddings generated by GCN and GCN$^+$ on Peptides-func.

Figure 5 shows the resulting t-SNE projections. We observe that the embeddings produced by GCN$^+$ exhibit more distinct class separation and greater inter-class margins compared to those produced by the vanilla GCN.

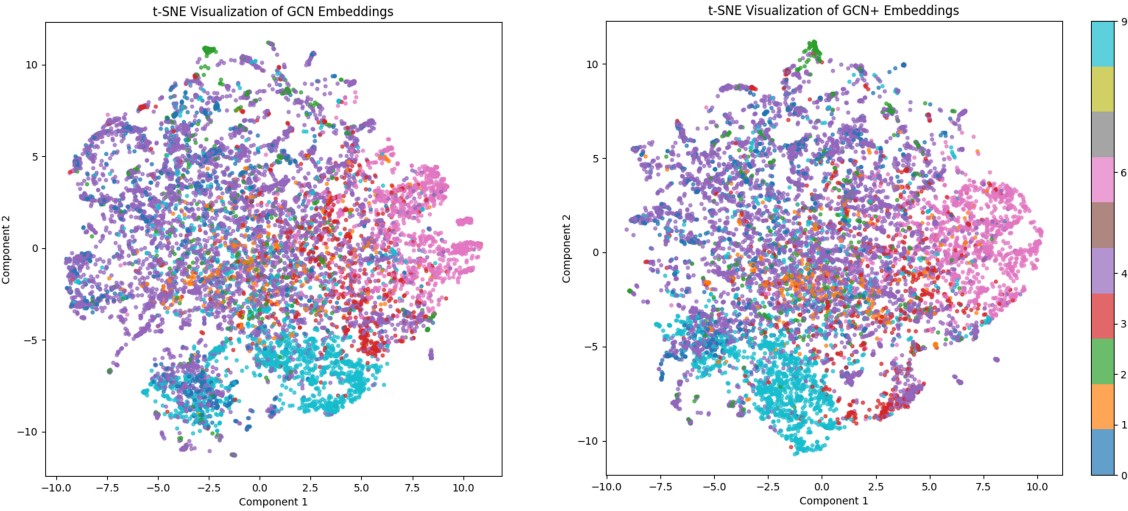

*Figure 5.* t-SNE visualization of graph-level embeddings learned by GCN (left) and GCN$^+$ (right). Colors denote different classes.

## B.5. Static Attention Problem of GTs

To gain insight into why GNNs can outperform GTs in graph-level tasks, we analyze the attention patterns in several representative GT architectures. We identified a potential limitation, which we call the *static attention problem*: the attention scores appear to be globally consistent across all nodes in the graph, irrespective of the query node.

As a concrete example, we analyze GraphGPS on a misclassified molecule from the ZINC dataset (Figure 6). As illustrated, nearly all nodes primarily focus on two structures: the five-membered pyrazoline ring in the top right and the benzene ring at the bottom. In contrast, functionally significant substructures, such as the C–O–N=N group in the top left and the nitro group (–NO2) in the bottom left, receive minimal attention. This query-invariant attention pattern results in insufficient sensitivity to subgraph structures, which adversely affects prediction accuracy.

In contrast, message-passing GNNs perform node-specific aggregation, allowing the model to capture diverse local substructures more effectively. This is beneficial for graph prediction tasks, where node representations are aggregated (e.g., via global pooling) into a global graph embedding. When nodes encode meaningful subgraph patterns, the resulting graph representation becomes more informative and discriminative.

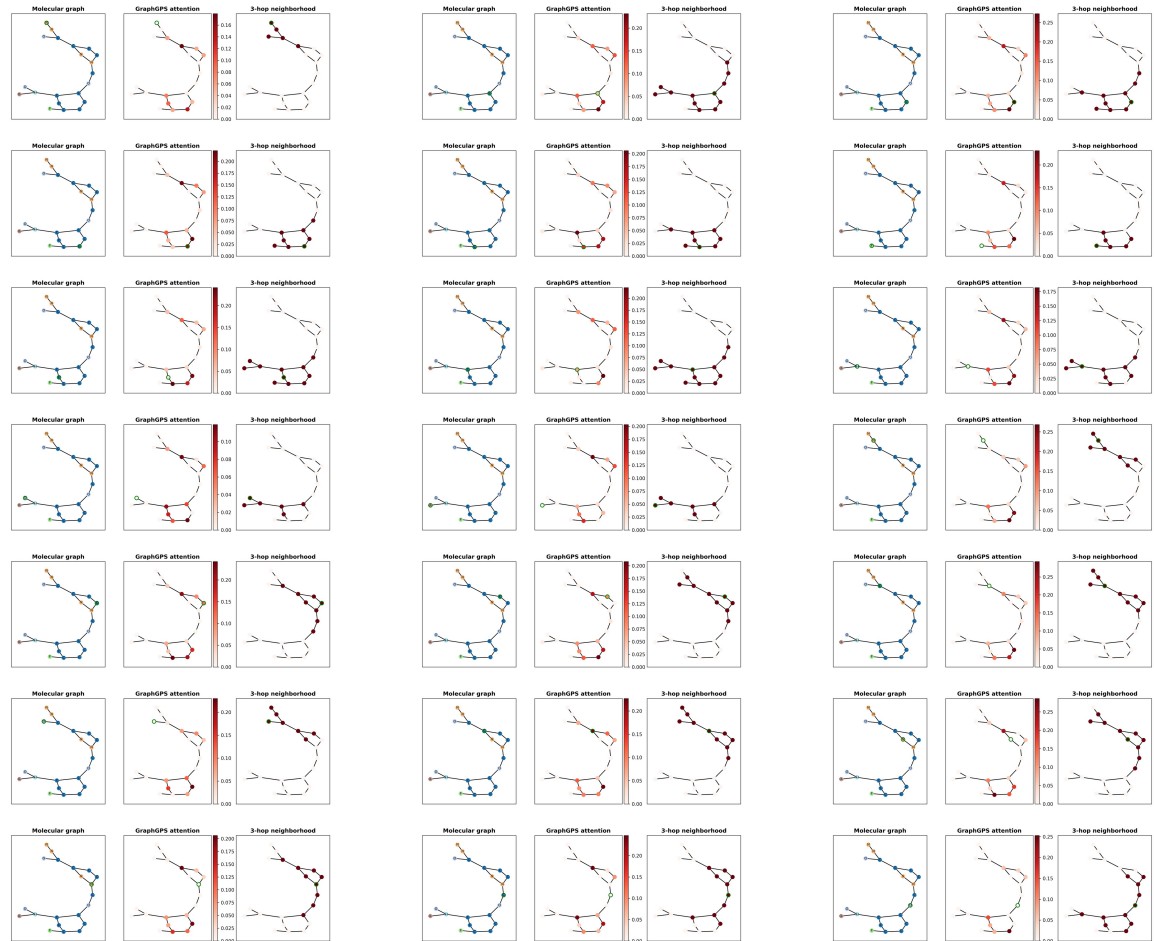

*Figure 6.* Visualization of GraphGPS attention scores across all nodes within the same molecular graph. The left side illustrates the molecular structure. The center column displays the attention scores of a node (highlighted with green borders) learned by the classic GraphGPS, while the right column showcases the 3-hop neighborhood of the node. Notably, the majority of nodes attend predominantly to two structural motifs: the five-membered ring (pyrazoline, top right) and the benzene ring (bottom). In contrast, functionally significant substructures—such as the C–O–N=N group (top left) and the nitro group (–NO2, bottom left)—receive minimal attention.

## C. Further Discussion

### C.1. Clarification on GNN$^+$ and Classic GNNs

GNN$^+$ enhances classic message-passing GNNs by integrating widely adopted architectural techniques including edge features, normalization, dropout, residual connections, FFN, and positional encoding. These components are not novel or specific to any one architecture; rather, they have been widely used across the literature on classic GNNs:

- **Residual connections**, **dropout**, and **normalization** were included in the original GCN (Kipf & Welling, 2017).
- **Edge feature** is already incorporated in the early general MPNN framework (Gilmer et al., 2017), which forms the foundation of many classic GNNs such as GatedGCN (Bresson & Laurent, 2017).
- The use of an **MLP/FFN** after message passing was introduced in the GIN paper (Xu et al., 2018).
- **Positional encoding** has been employed in prior GNN works (Dwivedi et al., 2021).

GNN$^+$ unifies these standard components within a consistent framework while preserving the core message-passing mechanism. Our results show that classic GNNs, when equipped with such enhancements, can achieve performance comparable to SOTA models on graph-level tasks. We view GNN$^+$ as a practical and faithful extension that unlocks the potential of classic GNNs. In this sense, GNN$^+$ is not a departure from the message-passing paradigm, but rather a natural evolution that reinforces its core strengths.

