# OpenReview forum: "Can Classic GNNs Be Strong Baselines for Graph-level Tasks? Simple Architectures Meet Excellence"
_ICML.cc/2025/Conference — ICML 2025 poster_

### Official Review · Reviewer_FRSj · 2025-02-22

**Overall Recommendation:** 2

**Summary:**

The paper presents GNN+, a framework that enhances Graph Neural Networks (GNNs) using six components—edge features, normalization, dropout, residual connections, feed-forward networks (FFNs), and positional encoding—to address issues such as over-smoothing and capturing long-range dependencies.

Through benchmark evaluations, GNN+ models demonstrate superior performance and efficiency compared to Graph Transformers (GTs), often securing top positions.

Each component of GNN+ significantly enhances the capability of GNNs, making them a simpler yet competitive option for graph-level tasks.

**Claims And Evidence:**

The claims are supported by empirical evidence, comparing GNN+ with state-of-the-art models across 14 datasets, as detailed in Tables 2, 3, and 4.

Additionally, thorough ablation studies isolate and evaluate the contributions of each component within the GNN+ framework in Tables 5 and 6.

However, the enhanced performance of GNN+ might depend heavily on meticulous hyperparameter tuning, suggesting that improvements might stem more from this tuning rather than the architecture's inherent superiority.

**Essential References Not Discussed:**

Previous research [2] highlights the necessity of standardizing experimental protocols in evaluating GNNs, as it reveals that many GNN architectures fail to consistently surpass structure-agnostic baselines, especially in chemical datasets.

This context is significant for GNN+ of this submission, as it aligns with rigorous evaluation standards, ensuring that its performance gains are genuine and robust rather than results of experimental artifacts or overfitting.

A detailed discussion of simple approaches [3, 4] would not only highlight alternative pathways to simplicity and efficiency but also provide valuable insights for further enhancing the design of classic GNNs as pursued in the GNN+ paper.

[2] A Fair Comparison of Graph Neural Networks for Graph Classification, In ICLR'20,
[3] A Simple yet Effective Method for Graph Classification, In IJCAI'22,
[4] A simple yet effective baseline for non-attributed graph classification, In ICLR'19 Workshop: Representation Learning on Graphs and Manifolds.

**Experimental Designs Or Analyses:**

The authors evaluated GNN+ across 14 benchmark datasets from three prominent sources (GNN Benchmark, LRGB, and OGB), reporting mean performance and standard deviations over five runs.

Moreover, the design includes comprehensive ablation studies in Section 5.2 that systematically remove individual components to isolate their contributions.

The hyperparameters used are reported in the Appendix Section A.3.

**Methods And Evaluation Criteria:**

The proposed methods and evaluation criteria are well aligned with the problem of improving graph-level performance.

By enhancing classic GNNs with six components, the approach directly addresses known limitations of traditional GNNs.

Moreover, the choice of benchmark datasets—including those from the GNN Benchmark, Long-Range Graph Benchmark (LRGB), and Open Graph Benchmark (OGB)—is appropriate because they cover diverse applications widely recognized in the community.

**Other Comments Or Suggestions:**

The submission would benefit from a discussion on limitations and future research directions.

For instance, despite the strong empirical results, a detailed theoretical analysis of when and why GNN+ surpasses the performance of Graph Transformers would enhance understanding.

Additionally, establishing formal bounds on expressiveness and generalization could inform the design of future models.

**Other Strengths And Weaknesses:**

Strengths:

[+] Besides achieving competitive or superior accuracy compared to state-of-the-art Graph Transformers, the paper demonstrates that the enhanced classic GNNs are computationally more efficient, an important practical advantage.

[+] The paper clearly outlines each component of the proposed method and provides well-structured experimental results. The detailed description of the experimental setup and hyperparameter tuning further enhances transparency and reproducibility.

Weaknesses:

[-] While the paper excels in its empirical contributions, it does not provide new theoretical insights or rigorous proofs to support the improvements.

[-] The performance gains appear to rely on extensive and careful hyperparameter tuning. This raises questions about the robustness of the improvements when applied in different settings or on different types of graphs beyond the benchmark datasets used.

**Questions For Authors:**

1. Was a hyperparameter sensitivity analysis performed for critical hyperparameters in GNN+, and if so, how sensitive is GNN+ to changes in these parameters?
2. Were all the baseline models, particularly the graph transformer baselines, tuned with the same level of rigor as GNN+?
3. Were the six components—edge features, normalization, dropout, residual connections, feed-forward networks (FFNs), and positional encoding—carefully integrated into the baseline models and compared with GNN+ models?
4. Was a trade-off analysis conducted to compare test data performance (e.g., accuracy) and training performance (e.g., training time) between fast baseline models and GNN+ models?

**Relation To Broader Scientific Literature:**

The key idea of the paper is to enhance classic GNNs by systematically integrating six techniques that are themselves well-established in the literature.

Four of these techniques—normalization, dropout, residual connections, and feed-forward networks—are well-recognised in the literature for improving GNN performance, albeit in node classification tasks [1].

The integration of the remaining two components—edge features and positional encoding—is straightforward, aligning with existing knowledge: edge features are crucial for molecular datasets, and positional encodings are widely used in graph transformers.

[1] Classic GNNs are Strong Baselines: Reassessing GNNs for Node Classification, In NeurIPS'24.

**Theoretical Claims:**

The submission does not include any formal theoretical proofs for its claims.

Instead, it presents standard equations that define classic GNN operations (such as those for GCN, GIN, and GatedGCN) along with modifications that integrate components such as edge feature integration, normalization, dropout, residual connections, FFN, and positional encoding.

These equations are largely descriptive and serve to illustrate how the GNN+ framework is constructed rather than providing rigorous proofs of new theoretical properties.

---

> ### Author Rebuttal · Authors · 2025-04-01
>
> Thank you for your thorough review and for acknowledging our contributions. We sincerely hope our response below further strengthens your confidence in our work.
>
> **(1) Related Works**
>
> Thank you for sharing these related works. We have integrated detailed discussions of these relevant studies ([2], [3], [4], and others) into the revised manuscript to better contextualize our contributions.
>
> **(2) Performance vs. Efficiency Trade-off Analysis of Fast Baseline Models and GNN+**
>
> Thank you for your thoughtful question. The GT baselines used global attention mechanisms with *quadratic* computational complexity, resulting in higher computational costs without performance gains compared to GNNs. To address your question, we conducted additional experiments using two *linear* baseline models—HRN [3] and LDP [4]—and compared their performance with GNN+.
>
> First, we evaluated these models on five **small-scale** graph classification datasets (IMDB-B, IMDB-M, COLLAB, MUTAG, and PTC) used in [3, 4]. These datasets consist of only 300–5000 graphs, substantially smaller than the main datasets (>10,000 graphs) used in our benchmarking study. Under identical experimental setups from their papers, GCN+ achieved comparable performance to these fast baselines. Given the small size of the datasets, their computation times are equally fast.
>
> |Model (Accuracy)|IMDB-B↑|IMDB-M↑|COLLAB↑|MUTAG↑|PTC↑|
> |-|-|-|-|-|-|
> |\# graphs|1000|1500|5000|188|344|
> |LDP|75.4|50.0|78.1|90.3|64.5|
> |HRN|77.5|52.8|81.8|90.4|65.7|
> |GCN+|76.9|52.3|80.9|90.1|66.6|
>
> Further, we extended this comparison to **large-scale** OGB datasets (ogbg-molhiv and ogbg-molpcba) using identical experimental setups. Although LDP and HRN showed faster training times than GCN+, their performance was significantly lower. This highlights the importance of rigorous benchmarking on larger scales. Evaluating models solely on small datasets, like MUTAG and PTC, can mask true differences in generalization capabilities. Results from large-scale datasets clearly demonstrate GCN+'s superior predictive performance, validating our focus on comprehensive and systematic benchmarking.
>
> |Model|ogbg-molhiv (41,127 graphs)||ogbg-molpcba (437,929 graphs)||
> |-|-|-|-|-|
> ||AUROC↑|Training Time (epoch/s)↓|Avg. Precision↑|Training Time (epoch/s)↓|
> |LDP|0.7121±0.0105|5s|0.1243±0.0031|31s|
> |HRN|0.7587±0.0147|9s|0.2274±0.0043|68s|
> |GCN+|0.8012±0.0124|16s|0.2721±0.0046|91s|
>
> Thank you once again for helping us improve our work. These results have been included in the revised version of our manuscript.
>
> **(3) Future Theoretical Research**
>
> We appreciate your comment and suggestion. We have incorporated a discussion of **limitations and future research directions** into the revised manuscript.
>
> Our work is an **empirical benchmarking study** (please see our reply to Reviewer s2AU "(1) Theoretical Analysis"). While deeper theoretical exploration would be valuable, such analysis lies **beyond the scope of this benchmarking study**.
>
> As noted by Reviewer 5voA, "*While the paper does not introduce new theoretical innovations, it effectively synthesizes existing research, providing valuable insights and a comprehensive summary of current knowledge in the field.*"
>
> The **empirical insights** derived from this work, particularly from the ablation studies (Tables 5 and 6), offer **valuable guidelines** for researchers in designing and applying GNNs to graph-level problems. These findings also **lay a solid foundation for future theoretical investigations**.
>
> **(4) Hyperparameter Tuning and Fairness of Comparison between GTs and GNNs**
>
> Thank you for highlighting this concern. We applied equally thorough hyperparameter tuning to all GT baselines, incorporating edge features, normalization, dropout, residual connections, FFNs, and positional encoding. All models were retrained using the same hyperparameter search space as GNN+ (lines 251–255).
>
> Notably, GNN+ remains robust and dataset-agnostic. Please see our response to TFXX, "(2) Fair Comparison between GTs and GNNs".
>
> **(5) Hyperparameter Sensitivity Analysis**
>
> Thank you for your thoughtful suggestion. In response, we conducted additional experiments to systematically evaluate the sensitivity of GNN+ performance to critical hyperparameters, specifically dropout rates ([link](https://anonymous.4open.science/r/GNN-3221/dropout.jpg)) and the number of layers ([link](https://anonymous.4open.science/r/GNN-3221/layer.jpg)).
>
> - **Dropout Rates:** Our results indicate that a low dropout rate (≤ 0.2) is sufficient and optimal, whereas higher dropout rates significantly degrade performance.
> - **Number of Layers:** Residual connections enable GNN+ to achieve optimal performance across a wide range. Increasing the number of layers does not lead to sudden performance drops, indicating effective mitigation of the over-smoothing problem.
>
> For other hyperparameters that are binary (enabled or disabled), please refer to our ablation studies in Tables 5 and 6.

---

### Official Review · Reviewer_TFXX · 2025-02-28

**Overall Recommendation:** 2

**Summary:**

This paper explores techniques inspired by Graph Transformers (GTs) to enhance Graph Neural Networks (GNNs). The authors demonstrate that these enhanced GNNs outperform most GTs on graph-level benchmarks, which contrasts with previous findings in the literature. Additionally, they provide empirical insights into the specific types of graphs on which each technique is most effective.

**Claims And Evidence:**

Yes.

**Essential References Not Discussed:**

No.

**Experimental Designs Or Analyses:**

Yes.

**Methods And Evaluation Criteria:**

See **Other Strengths And Weaknesses**.

**Other Comments Or Suggestions:**

In the ablation study section, I recommend emphasizing the analysis to ensure it isn't overshadowed by the description of the phenomena.

**Other Strengths And Weaknesses:**

**Strengths.** The experiment is comprehensive with respect to benchmarks and baselines. It effectively validates the argument that "GNNs with enhancements can surpass Graph Transformers on graph-level tasks," providing valuable insights for the literature.

**Weakness.**
-  The paper dedicates considerable space to experimental settings, results, and observations, but it lacks **deeper analysis**, which would offer more valuable and insightful contributions. The proposed method feels more like a successful technical trial. Notably, I’m not dismissing technical works; my main concern is that the finding—'GNNs with enhancements can surpass Graph Transformers on graph-level tasks'—**alone** is not enough to support a strong paper.
- I notice that the six techniques are selectively applied, as shown in Table 6. This implies that the **GNN+ architectures vary across benchmarks, with only the best-performing configurations being reported from Table 2 to Table 4**. In contrast, the architecture of the GT baselines remains consistent across benchmarks. This discrepancy introduces a degree of unfairness and weakens the argument that 'GNNs perform better than GT.' Additionally, the GNN+ architecture appears to be less practical due to the heavy dependence on the dataset.

**Questions For Authors:**

1. For the six techniques explored, there are **multiple ways to combine them** through operations like reordering, yet the GNN+ formulation is fixed as in Equation 11.  Other similar confusion occurs—why choose BN over LN, and why add edge features in the chosen way? ... Do other variants perform similarly well or worse. If they perform worse, are there any explanations? Additionally, it would be valuable to provide insights into **how we should enhance GNNs for graph-level tasks**.


2. I note that GTs also incorporate all the techniques including edge feature integration, normalization, dropout, residual connections, feed-forward networks, and positional encoding. **It seems that the attention machenism may not offer benefits, and may even hurt performance compared to message-passing.** Are there any explanations for this observation?

**Relation To Broader Scientific Literature:**

The findings prompt a reevaluation of whether Graph Transformers are truly necessary, given their complexity, especially considering that enhanced GNNs can achieve superior performance on graph-level tasks.

**Theoretical Claims:**

No proofs to be checked.

---

> ### Author Rebuttal · Authors · 2025-04-01
>
> Thank you for recognizing the depth of our experiments and insights. We believe some key points may have been missed and hope our clarification encourages you to revisit and re-evaluate our work.
>
> **(1) Deeper Theoretical Analysis**
>
> Thank you for your valuable comment. Please refer to our response to Reviewer s2AU, "(1) Theoretical Analysis".
>
> **(2) Fair Comparison between GTs and GNNs**
>
> We'd like to clarify that our performance comparison is **fair**.
>
> Firstly, the techniques used for GNN+, regarded as hyperparameters—including edge feature module, normalization, dropout, residual connections, FFNs, and positional encoding—have already been incorporated into all GT baselines. Importantly, we retrained the GT baselines using **the same hyperparameter search space as the classic GNNs** (lines 251–255). Therefore, the application of these techniques is **consistent** across both GNN+ and GT baselines.
>
> Secondly, to comprehensively address your concern, we conducted additional experiments utilizing **a fixed model with all six techniques integrated**, denoted as **GNN+ (fixed)**, as presented in the table below. The results show that the fixed GNN+ model achieves performance comparable to the best configurations reported in the paper, demonstrating its practicality and robustness.
>
> |Model|PascalVOC-SP↑|COCO-SP↑|MalNet-Tiny↑|ogbg-molpcba↑|ogbg-code2↑|MNIST↑|CIFAR10↑|PATTERN↑|CLUSTER↑|
> |-|-|-|-|-|-|-|-|-|-|
> |**GCN+**|0.3357±0.0087|0.2733±0.0041|0.9354±0.0045|0.2721±0.0046|0.1787±0.0026|98.382±0.095|69.824±0.413|87.021±0.095|77.109±0.872|
> |**GCN+ (fixed)**|0.3341±0.0055|0.2716±0.0034|0.9235±0.0060|0.2694±0.0059|0.1784±0.0029|98.257±0.063|69.436±0.265|87.021±0.095|76.352±0.757|
> |**GatedGCN**|0.4263±0.0057|0.3802±0.0015|0.9460±0.0057|0.2981±0.0024|0.1896±0.0024|98.712±0.137|77.218±0.381|87.029±0.037|79.128±0.235|
> |**GatedGCN (fixed)**|0.4204±0.0061|0.3774±0.0028|0.9450±0.0045|0.2981±0.0024|0.1889±0.0018|98.712±0.137|77.218±0.381|87.029±0.037|79.128±0.235|
>
> **(3) Justification for Architectural Choices**
>
> > There are multiple ways to combine six techniques through operations like reordering, yet the GNN+ formulation is fixed as in Equation 11.
>
> Indeed, the GNN+ framework **is not limited to the formulation shown in Equation 11**. Instead, it represents a flexible GNN architecture that integrates edge features, normalization, dropout, residual connections, FFNs, and positional encoding, which collectively define GNN+. Each component allows various implementation choices; for instance, both BN and LN are viable options for normalization.
>
> The specific combination presented in Equation 11 was chosen because our experiments consistently showed strong and, in some cases, remarkable performance. However, our intention was not to advocate exclusively for this configuration. Rather, our primary goal was to clearly demonstrate, through systematic empirical benchmarking, that an enhanced GNN architecture can match or surpass GTs across diverse datasets. Future research can explore alternative architectures and refinements within this flexible framework.
>
> **(4) Insights into How to Enhance GNNs for Graph-Level Tasks**
>
> Our ablation studies (Tables 5 and 6) provide detailed, actionable insights. For example, normalization substantially impacts larger-scale datasets but has less effect on smaller datasets, and a low dropout rate (≤0.2) consistently proves optimal. These findings serve as practical recommendations for researchers seeking to enhance GNN performance.
>
> **(5) Why Attention Fails**
>
> Thank you for the thought-provoking question. This phenomenon is the core finding of our study. While GTs employ the global attention mechanism, our results suggest that **it does not benefit graph-level learning as expected** and may even degrade performance by introducing unnecessary complexity.
>
> To investigate this, we visualized the attention scores of GraphGPS on the Peptides-func dataset (see [this link](https://anonymous.4open.science/r/GNN-3221/attention.jpg)). The visualization shows that the nodes in question (highlighted with green borders) predominantly attend to only one or a few distant nodes, or randomly attend to multiple distant nodes without clear, explainable patterns.
>
> An ablation study is presented in the table below. The results clearly indicate that the global attention mechanism used in GraphGPS may negatively impact performance. We hypothesize that the global attention mechanism conflicts with the local message-passing mechanism of GNNs, causing the model to excessively attend to distant, less relevant nodes. This phenomenon, referred to as the **over-globalizing problem**, has also been experimentally validated in recent studies [1].
>
> |Model|Peptides-func↑|
> |-|-|
> |**GraphGPS (GNN + Attention)**|0.6534 ± 0.0091|
> |**GraphGPS (GNN only, w/o Attention)**|0.6951 ± 0.0134|
> |**GraphGPS (Attention only, w/o GNN)**|0.6366 ± 0.0163|
>
> [1] Less is More: On the Over-Globalizing Problem in Graph Transformer, ICML 2024.

---

> > ### Comment · Reviewer_TFXX · 2025-04-02
> >
> > Thanks for your careful response. Some key questions still remain for me:
> >
> > 1. **A specific GNN architecture is not enough to support the claim that “classic GNNs excel in graph-level tasks”.** Given that GNN+ is dedicatedly designed with many tricks (different from may classic GNN models), it may be questionable whether another different GNN architecture has the same effect. Crucially, GNN+ only replaces the attention mechanism of GTs with message passing, and the other parts “*including edge feature integration, normalization, dropout, residual connections, FFN, and PE—is indispensable*”. **I think the resulting architecture can not be claimed as “classic GNNs”.**
> > 2. For me, explaining why “GNNs can surpass GTs in graph-level tasks” will be necessary and intriguing. However, this is not explicitly discussed in the paper. **In contrast, it seems that the *over-globalizing* problem revealed in [1] is actually the key insight.**
> >
> > I really recognize your contributions on experimental validations and technical recommendations, and will raise my score accordingly. However, my concern remains that these can not support a general conclusion “classic GNNs meet excellence” and lack novel intriguing insights.

---

> > > ### Author Response · Authors · 2025-04-04
> > >
> > > Dear Reviewer TFXX,
> > >
> > > Thank you for your further feedback and for raising your score!
> > >
> > > **(1) The resulting architecture GNN+ can not be claimed as "classic GNNs"**
> > >
> > > Thank you for your thoughtful comment.
> > >
> > > We'd like to emphasize that the **main difference between GTs and GNNs lies in their core mechanisms: global attention vs. message-passing**. Other techniques, such as edge feature integration, normalization, dropout, residual connections, FFN, and PE, are standard practices that are widely adopted across various network architectures and can be flexibly integrated. These techniques are commonly found in the literature on classic message-passing GNNs:
> > >
> > > 1. **Residual connections**, **dropout**, and **normalization** were already utilized in the original **GCN** paper (Kipf & Wellings, 2017).
> > > 2. **Edge feature** is already incorporated in the early general MPNN framework (Gilmer et al., 2017), which forms the foundation of many classic GNNs such as **GatedGCN** (Bresson & Laurent, 2017).
> > > 3. The use of an **MLP/FFN** after message passing was introduced in the **GIN** paper (Xu et al., 2018).
> > > 4. **PE** has been employed in prior GNN works (Dwivedi et al., 2021).
> > >
> > > Our GNN+ integrates these widely-used techniques into a unified framework that encompasses various classic GNNs, such as GCN, GIN, and GatedGCN, to enhance their performance. Extensive experiments demonstrate that these classic GNN models, when enhanced by the GNN+ architecture, excel at graph-level tasks. Thus, GNN+ is an effective architecture that unlocks the potential of classic GNNs, though it may not be the only one. In this context, we believe it is fair to say that classic GNNs excel at graph-level tasks.
> > >
> > > **(2) Why GNNs can surpass GTs in graph-level tasks**
> > >
> > > To explore why "GNNs can surpass GTs in graph-level tasks", we conducted a thorough analysis of attention mechanisms within GTs, uncovering a critical problem we term the **static attention problem**: the attention scores in GTs seem to be globally consistent across all nodes in the graph, regardless of the query node. We present visualizations ([link](https://anonymous.4open.science/r/attn-4558/README.md)) of the attention scores across multiple datasets, including ZINC, Peptides-func, Peptides-struct, MNIST, ogbg-molhiv, PascalVOC-SP, PATTERN, CLUSTER, and CIFAR10. The results indicate that a small set of nodes consistently receives dominant attention across all nodes, restricting the model's ability to focus on task-relevant localized structures. It's important to note that the visualized graphs were randomly selected, and this pattern consistently appears across different layers.
> > >
> > > For example, we analyze GraphGPS attention scores on a misclassified molecule from the ZINC dataset, which is ideal for interpretability due to its small graph size ([link to visualization](https://anonymous.4open.science/r/attn-4558/zinc_example.jpg)). As illustrated, nearly all nodes primarily focus on two structures: the five-membered pyrazoline ring in the top right and the benzene ring at the bottom. In contrast, functionally significant substructures, such as the C–O–N=N group in the top left and the nitro group (–NO₂) in the bottom left, receive minimal attention. This query-invariant attention pattern results in insufficient sensitivity to subgraph structures, which adversely affects prediction accuracy.
> > >
> > > In contrast, message-passing GNNs perform node-specific aggregation, allowing the model to capture diverse local substructures more effectively. This is beneficial for graph prediction tasks, where node representations are aggregated (e.g., via global pooling) into a global graph embedding. When nodes encode meaningful subgraph patterns, the resulting graph representation becomes more informative and discriminative.
> > >
> > > We further validate this through an ablation study on the ZINC dataset below.
> > >
> > > |Model|ZINC↓|
> > > |-|-|
> > > |**GraphGPS (GNN + Attention)**|0.070|
> > > |**GraphGPS (GNN only)**|0.070|
> > > |**GraphGPS (Attention only)**|0.217|
> > >
> > > Moreover, the static attention problem contributes to over-smoothing, wherein similar attention patterns yield near-identical node embeddings across the graph. We investigated this by comparing the Dirichlet energy of GraphGPS with GCN+ across four datasets. GraphGPS consistently showed lower Dirichlet energy, indicating reduced node representation diversity due to static attention, which further diminishes performance.
> > >
> > > |Model (Dirichlet energy)|Peptides-func↑|CLUSTER↑|MNIST↑|CIFAR↑|ZINC↑|
> > > |-|-|-|-|-|-|
> > > |**GraphGPS**|32.233|0.256|8.376|5.637|2.679|
> > > |**GCN+**|**80.506**|**0.624**|**21.127**|**11.582**|**3.966**|
> > >
> > > These results support our observation: the global attention mechanism in GTs not only suffers from the *over-globalizing problem* identified in [1] but also exhibits a **static attention problem**. In contrast, GNNs effectively capture node-dependent subgraph patterns, which is a key reason they can outperform GTs in graph-level tasks.
> > >
> > > Best Regards,
> > >
> > > The Authors

---

### Official Review · Reviewer_5voA · 2025-03-09

**Overall Recommendation:** 3

**Summary:**

This study explores the potential of Graph Neural Networks (GNNs) by enhancing them with the GNN+ framework, which incorporates techniques such as edge feature integration, normalization, and positional encoding. The results show that classic GNNs, enhanced with GNN+, outperform Graph Transformers (GTs) on graph-level tasks, achieving top rankings across 14 datasets. While the paper does not present theoretical innovations, it provides an insightful summary and practical evaluation of existing methods, challenging the notion that complex Graph Transformers are necessary for superior performance.

**Claims And Evidence:**

Yes

**Essential References Not Discussed:**

No

**Experimental Designs Or Analyses:**

Yes

**Methods And Evaluation Criteria:**

Yes

**Other Comments Or Suggestions:**

1. To enhance your paper's contributions, it is recommended to incorporate and compare your proposed method with state-of-the-art techniques addressing over-smoothing and over-squashing in Graph Neural Networks (GNNs) (you mentioned in the first sentence of the Abstract), such as Multi-Track Message Passing and Cooperative Graph Neural Networks. Conducting comparative experiments with these methods will provide a comprehensive evaluation of your approach's effectiveness and highlight its contributions to the field.

2. Additionally, performing experiments that involve increasing the number of network layers to observe performance changes can help assess and address the over-smoothing issue in GNNs.

**Other Strengths And Weaknesses:**

While the paper does not introduce new theoretical innovations, it effectively synthesizes existing research, providing valuable insights and a comprehensive summary of current knowledge in the field.

**Questions For Authors:**

Why doesn't the author focus on graph node classification instead of graph classification?

**Relation To Broader Scientific Literature:**

This study challenges the prevailing belief that complex Graph Transformer (GT) mechanisms are essential for superior graph-level performance by demonstrating that enhanced classic Graph Neural Networks (GNNs), utilizing the GNN+ framework, can achieve top rankings across multiple datasets.

**Theoretical Claims:**

Yes

---

> ### Author Rebuttal · Authors · 2025-04-01
>
> We greatly appreciate the very detailed feedback and your recognition of our contributions! We hope our response below will further enhance your confidence in our work.
>
> **(1) Comparison with SOTA GNNs Addressing Over-smoothing and Over-squashing**
>
> > To enhance your paper's contributions, it is recommended to incorporate and compare your proposed method with state-of-the-art techniques addressing over-smoothing and over-squashing in Graph Neural Networks (GNNs) (you mentioned in the first sentence of the Abstract), such as Multi-Track Message Passing and Cooperative Graph Neural Networks. Conducting comparative experiments with these methods will provide a comprehensive evaluation of your approach's effectiveness and highlight its contributions to the field.
>
> We appreciate the thoughtful suggestion. Following your recommendation, we conducted additional experiments on the Peptides-func and Peptides-struct datasets, comparing GCN+ with SOTA GNNs including Multi-Track Message Passing (MTGCN) [1] and Cooperative Graph Neural Networks (CO-GNN) [2]. These experiments adhered to the hyperparameter settings described in our paper (Lines 264-273).
>
> For CO-GNN, we tuned its message-passing mechanisms, including SUMGNNs, MEANGNNs, GCN, GIN, and GAT, as recommended by the original paper. For MTGCN, we optimized the number of message-passing stages (from 1 to 4) to ensure a fair comparison.
>
> The comparative results in the table below clearly demonstrate the effectiveness of our GNN+ framework in addressing over-smoothing and over-squashing. We have included the results in the revised version.
>
> | Model    | Peptides-func ↑     | Peptides-struct ↓   |
> | -------- | ------------------- | ------------------- |
> | MTGCN    | 0.6936 ± 0.0089     | 0.2461 ± 0.0019     |
> | CO-GNN   | 0.7012 ± 0.0106     | 0.2503 ± 0.0025     |
> | **GCN+** | **0.7261 ± 0.0067** | **0.2421 ± 0.0016** |
>
> **(2) The Impact of the Number of Network Layers on GNN+**
>
> > Additionally, performing experiments that involve increasing the number of network layers to observe performance changes can help assess and address the over-smoothing issue in GNNs.
>
> Thank you for the thoughtful suggestion. In response, we conducted additional experiments to examine the impact of the number of network layers on GNN+ for the PATTERN, CLUSTER, and PascalVOC-SP datasets. The detailed results can be found at this [link](https://anonymous.4open.science/r/GNN-3221/layer.jpg).
>
> Thanks to residual connections, GNN+ achieves optimal performance across a wide range of layers. Specifically, on PATTERN, both GNN+ variants (GCN+ and GatedGCN+) achieve optimal performance at 12 layers. In contrast, on CLUSTER and PascalVOC-SP, their performance continues to improve as the number of layers increases.
>
> Overall, **GNN+ maintains strong predictive performance even at greater depths**, demonstrating its ability to effectively mitigate the over-smoothing issue commonly observed in GNNs.
>
> **(3) Why Not Consider Graph Node Classification**
>
> > Why doesn't the author focus on graph node classification instead of graph classification?
>
> Thank you for your question. Our work is inspired by a recent study [3] that showed classic GNNs can achieve performance comparable to, or even surpassing, state-of-the-art GTs for node-level tasks, such as node classification. However, there has been no similar conclusion or investigation for graph-level tasks, and our work aims to fill this gap.
>
> Please note that we have addressed inductive node classification, which is considered one of the graph-level tasks. Specifically, Table 2 and 3 (in our original manuscript) include results on the PATTERN, CLUSTER, PascalVOC-SP, and COCO-SP datasets, which evaluate the performance of inductive node classification.
>
> In addition, although our GNN+ framework is specifically designed for graph-level tasks, it can also be applied to node-level tasks. Below are the node classification results on four datasets from CO-GNN: roman-empire, amazon-ratings, minesweeper, and questions. The results indicate that GCN+ consistently achieves performance comparable to CO-GNN.
>
> | Model  | roman-empire | amazon-ratings | minesweeper  | questions    |
> | ------ | ------------ | --------------- | ------------ | ------------ |
> |        | Accuracy↑    | Accuracy↑       | AUROC↑       | AUROC↑       |
> | GCN    | 73.69 ± 0.74 | 48.70 ± 0.63    | 89.75 ± 0.52 | 76.09 ± 1.27 |
> | CO-GNN | 91.57 ± 0.32 | 54.17 ± 0.37    | 97.31 ± 0.41 | 80.02 ± 0.86 |
> | GCN+   | 91.27 ± 0.20 | 53.80 ± 0.60    | 97.86 ± 0.24 | 79.02 ± 0.60 |
>
> ---
> [1] Multi-Track Message Passing: Tackling Oversmoothing and Oversquashing in Graph Learning via Preventing Heterophily Mixing, ICML 2024.
>
> [2] Cooperative Graph Neural Networks, ICML 2024.
>
> [3] Classic GNNs are Strong Baselines: Reassessing GNNs for Node Classification, NeurIPS 2024.

---

> > ### Comment · Reviewer_5voA · 2025-04-02
> >
> > Thanks for the author's reply. I'd like to maintain my score.

---

> > > ### Author Response · Authors · 2025-04-03
> > >
> > > Dear Reviewer 5voA,
> > >
> > > Thank you for taking the time to review our rebuttal. We highly appreciate your positive and insightful assessment of our work, as well as your reaffirmation of your rating.
> > >
> > > We are particularly grateful for your recognition that "while the paper does not present theoretical innovations, it provides an insightful summary and practical evaluation of existing methods, challenging the notion that complex Graph Transformers are necessary for superior performance." Your acknowledgment of the value of our empirical analysis and synthesis of prior work is truly encouraging and affirms the importance of reassessing basic models with careful benchmarking.
> > >
> > > In case there are any additional concerns we can address, please let us know.
> > >
> > > Best Regards,
> > >
> > > The Authors

---

### Official Review · Reviewer_s2AU · 2025-03-14

**Overall Recommendation:** 1

**Summary:**

The paper challenges the prevailing assumption that Graph Transformers are inherently superior to Message-Passing GNNs for graph-level tasks. It introduces GNN+, a framework enhancing three classic GNNs (GCN, GIN, GatedGCN) with six techniques: edge feature integration, normalization, dropout, residual connections, feed-forward networks (FFN), and positional encoding (RWSE). Evaluated across 14 graph-level datasets (GNN Benchmark, LRGB, OGB), GNN+ achieves top-three rankings on all datasets and first place on eight, outperforming SOTA GTs while being more computationally efficient. The results suggest that classic GNNs, when properly enhanced, are highly competitive for graph-level tasks, challenging the necessity of complex GT architectures.

**Claims And Evidence:**

The key claim of this paper is that classic GNNs can match the performance of GTs on graph-level tasks by incorporating certain tricks and conducting meticulous parameter search. This claim is supported by substantial empirical evidence. However, this claim holds limited value as it lacks relevant theoretical analysis and deeper mechanistic insights.

**Essential References Not Discussed:**

Key prior works are appropriately cited.

**Experimental Designs Or Analyses:**

I have checked the experiment section, which is quite comprehensive. They include 14 datasets spanning regression, classification, and inductive tasks and provide rigorous analysis of each component.

**Methods And Evaluation Criteria:**

This paper introduces GNN+, an architecture that augments classic GNNs with edge features, normalization, dropout, residual connections, FFNs, and PE. They evaluated GNN+ on 14 graph-level datasets. The evaluation criteria used make sense. However, the work introduces no novel methodological advancements, and the idea of tuning classic GNNs has already been proposed in prior literature.

**Other Comments Or Suggestions:**

I have no other comments or suggestions.

**Other Strengths And Weaknesses:**

This paper comprehensively re-examines the performance of GNNs on graph-level tasks, conducting extensive experiments across datasets of varying scales, which is particularly impressive. However, the work suffers from the following limitations:
1. The study primarily offers empirical observations rather than a deep mechanistic analysis to establish universal design principles for graph-level GNNs. Consequently, its utility in guiding researchers to design or apply GNNs for graph-level tasks remains constrained.
2. The core methodology (GNN+) constitutes a direct extension of Luo et al.'s framework for graph-level tasks, with insufficient technical novelty. Notably, while graph-level tasks fundamentally differ from node-level tasks, the paper fails to elucidate how the introduced tricks (e.g., residual connections, PE) specifically enhance graph-level capabilities, such as improving expressiveness for the graph isomorphism problem.
3. Despite reporting exhaustive quantitative results, the absence of visualizations hinders understanding of GNN+'s operational advantages and decision-making patterns.

**Questions For Authors:**

Refer to Weaknesses.

**Relation To Broader Scientific Literature:**

The paper directly challenges recent works advocating GTs (e.g., GraphGPS) by showing that classic GNNs remain competitive. It extends findings from Luo et al. (2024) to graph-level tasks.

**Theoretical Claims:**

The paper is empirical and does not propose new theoretical claims or proofs.

---

> ### Author Rebuttal · Authors · 2025-04-01
>
> Thank you for your constructive feedback. We believe there may have been some misinterpretations of our work that could have influenced your assessment. We hope our clarifications encourage you to reassess our work.
>
> **(1) Theoretical Analysis**
>
> > This claim holds limited value as it lacks relevant theoretical analysis and deeper mechanistic insights.
>
> We’d like to clarify that **our work is an empirical benchmarking study**, akin to previous notable benchmarking research such as [1,2,3,4,5]. Consequently, **theoretical analysis is neither intended nor within the scope of this work**.
>
> Furthermore, we'd like to explain **why the empirical study alone represents a significant contribution** to the community.
>
> In recent years, GTs have emerged as the leading approach for graph-level tasks, often dominating leaderboards, especially on small molecular graphs. This trend has fostered a growing perception that GTs, due to their global attention mechanisms, are inherently superior to GNNs, leading to the marginalization of GNNs in the field.
>
> However, our **comprehensive and fully reproducible experimental results**, as presented in Tables 2-4, provide compelling evidence that classic GNNs, when enhanced with our GNN+ framework, can consistently match or outperform GTs across a diverse suite of graph-level tasks.
>
> This finding has significant implications:
>
> 1. It suggests the **global attention** mechanisms of GTs **may not be as useful as commonly believed**. In fact, our ablation study shows they may degrade performance (see our response to Reviewer TFXX, "(5) Why Attention Fails").
>
> 2. It questions the need for complex architectures in graph-level tasks and **could prompt a methodological shift** from overly complex GTs to simpler GNN models, potentially reshaping the field's landscape.
>
> 3. It explains why state-of-the-art GTs often incorporate **message-passing mechanisms** into their models, either implicitly or explicitly, due to their **high effectiveness**.
>
> As noted by Reviewer 5voA, "*While the paper does not introduce new theoretical innovations, it effectively synthesizes existing research, providing valuable insights and a comprehensive summary of current knowledge in the field.*"
>
> [1] Pitfalls of Graph Neural Network Evaluation, NeurIPS 2018.
>
> [2] A Critical Look at the Evaluation of GNNs under Heterophily: Are We Really Making Progress, ICLR 2023.
>
> [3] Evaluating Graph Neural Networks for Link Prediction: Current Pitfalls and New Benchmarking, NeurIPS 2023.
>
> [4] A Fair Comparison of Graph Neural Networks for Graph Classification, ICLR 2020.
>
> [5] Classic GNNs are Strong Baselines: Reassessing GNNs for Node Classification, NeurIPS 2024.
>
> **(2) Insight into Why Certain Components Help**
>
> > The study's utility in guiding researchers to design or apply GNNs for graph-level tasks remains constrained.
>
> Our ablation studies (Tables 5 and 6) provide **empirical insights** into the contributions of each component of the proposed GNN+ architecture. For example, normalization has a more substantial impact on larger-scale datasets while being less pronounced on smaller ones; similarly, a very low dropout rate (≤0.2) consistently proves optimal for graph-level tasks. These findings greatly improve the usefulness of our work in helping researchers design and apply GNNs for graph-level problems. While we agree that deeper theoretical exploration would further benefit the community, such analysis lies beyond the scope of this benchmarking study. Nonetheless, **our empirical results lay a solid foundation for future theoretical investigations** into how these architectural components enhance graph-level expressiveness and effectiveness.
>
> **(3) Methodological Novelty**
>
> > The core methodology (GNN+) constitutes a direct extension of Luo et al.'s framework for graph-level tasks.
>
> It is important to note that GNN+ is **NOT** a trivial extension of Luo et al.'s model, which is intended for node-level tasks. Graph-level tasks, especially those involving small-scale molecular graphs, pose unique challenges. To address these, GNN+ integrates various techniques specifically tailored for graph-level modeling, such as edge feature module, FFNs and PE. The simple yet effective design of GNN+ was **established through extensive experiments conducted over half a year**, representing a new framework in the field.
>
> **(4) Visualization Results**
>
> Thank you for the suggestion. Following the advice, we have added additional visual analyses (e.g., t-SNE plots of learned embeddings ([link](https://anonymous.4open.science/r/GNN-3221/tsne.jpg)), sensitivity analysis of network depth ([link](https://anonymous.4open.science/r/GNN-3221/layer.jpg)) and dropout rate ([link](https://anonymous.4open.science/r/GNN-3221/dropout.jpg))). In the t-SNE figure, we observe that the graph embeddings generated by GCN+ exhibit greater inter-class distances compared to those generated by GCN. We have incorporated the results in the revised version.

---

> > ### Comment · Reviewer_s2AU · 2025-04-05
> >
> > Thank you for the authors' response. I appreciate the comprehensive experiments and analyses presented in this paper, but my primary concerns remain unaddressed:
> > 1. I acknowledge that the experiments provide empirical insights into the contributions of individual components, and I am not demanding rigorous theoretical proofs. The critical issue lies in whether this work offers sufficiently profound guidance for GNN design in graph-level tasks. The current study only presents a limited perspective: "equipping classical GNNs with specific techniques (normalization, dropout, edge features, residual connections, FFNs, positional encodings) improves performance." I agree with Reviewer TFXX’s critique: "I’m not dismissing technical works; my main concern is that the finding alone is not enough to support a strong paper."
> > 2. As previously noted, this work resembles an extension of Luo et al. (2024) to new datasets rather than a principled investigation tailored to graph-level challenges. The authors claim to design a novel GNN framework specifically for graph-level tasks, yet the added components exhibit no inherent graph-level specificity, as they are also applicable to node-level tasks. GNNs for graph-level prediction fundamentally require comparable representations across irregular graphs with diverse sizes, which involves critical problems on GNNs' expressiveness, graph isomorphism, and so on. Unfortunately, the empirical analyses in this work offer limited insights into these core graph-level challenges.
> >
> > Therefore, I would keep my rating.

---

> > > ### Author Response · Authors · 2025-04-05
> > >
> > > Dear Reviewer s2AU,
> > >
> > >
> > > Thank you for your further feedback.
> > >
> > > We regret that our work may not have aligned with your expectations, *despite our clarification that it is an **empirical benchmarking study** rather than a proposal of novel GNN models*. It appears there may be a **misunderstanding** regarding the **focus of our work**.
> > >
> > > Benchmarking studies are a crucial component of machine learning research, providing a foundational basis for advancing the field. Their importance is increasingly recognized by leading machine learning conferences, some of which feature **dedicated tracks for benchmarking studies**.
> > >
> > >
> > > > The authors claim to design a novel GNN framework specifically for graph-level tasks, yet the added components exhibit no inherent graph-level specificity, as they are also applicable to node-level tasks.
> > > >
> > > > GNNs for graph-level prediction fundamentally require comparable representations across irregular graphs with diverse sizes, which involves critical problems on GNNs' expressiveness, graph isomorphism, and so on. Unfortunately, the empirical analyses in this work offer limited insights into these core graph-level challenges.
> > >
> > > Our **GNN+** framework is specifically **designed to benchmark classic GNN models** for graph-level tasks, with the goal of **understanding their potential and promoting empirical rigor**. While architectural innovations and theoretical results—involving expressiveness and graph isomorphism—are important, they *fall outside the scope of our empirical study*.
> > >
> > > > The current study only presents a limited perspective: equipping classic GNNs with specific techniques improves performance.
> > >
> > > Our benchmarking study rigorously evaluates **over 30 state-of-the-art models** published in the past three years at top-tier machine learning conferences, focusing on graph-level tasks. The results, derived from **extensive experiments conducted over six months**, offer a **valuable resource for future research**. Importantly, our study reveals a **key finding**: *simple GNN models achieve state-of-the-art performance on graph-level tasks, indicating that complex graph transformers and their attention mechanisms may not be necessary.*
> > >
> > > We kindly request that you **evaluate this paper as a benchmarking study**, focusing on its strengths and weaknesses within that context, rather than as a proposal for new GNN models.
> > >
> > > Thank you.
> > >
> > > The Authors

---

### Decision · Program_Chairs · 2025-05-01

**Decision:**

Accept (poster)

**Comment:**

The paper argues that by enhancing them with edge feature integration, normalization, dropout, residual connections, feed-forward networks (FFN), and positional encoding, classic (message passing-based) GNNs match or surpass the performances of graph transformers, contrary to the common belief. This hypothesis is tested (and largely confirmed) in 14 datasets and against 21 graph transformer architectures, in what appears to be a fair comparison.

The paper was welcomed with mixed reviews. Most of the concerns can be summarized as follows:
- the paper does not provide theoretical insights
- it does not offer design insights
- the finding is over-generalized

Personally, all these concerns appear inconsistent, especially after the rebuttal.

- First of all, this is a benchmark paper whose focus is not on theory. Clearly, theory would add further value to this paper; however, theoretical insights were not central in studies which share the same objective as this [1-4]. In this sense, I **expect** the authors to emphasize the empirical nature of the insights offered by this paper in their camera-ready version, and to explicitly acknowledge the lack of a theoretical analysis as one of its current limitations (which should be addressed by subsequent works).
- The paper **does** offer design insights: arguably, the core contribution of the paper is a specific design which makes classic message-passing architectures compete with graph transformers, while being more efficient. Its primary contribution is indeed a way to design a GNN architecture which empirically matches or surpasses GTs' performances, where the importance of each component to the final architecture is tested via ablation studies.
- The main finding is shown on three different GNN architectures compared to a whole lot of graph transformers in a wide range of datasets. Given the width and depth of the analysis, I see a clear trend rather than an over-generalization.

Overall, the findings in this article are also valuable for GNN practitioners, which could, e.g., decide to resort the proposed architecture rather than GTs when efficiency is critical.

In my opinion, the graph learning community has only to benefit from these benchmarking analyses that test very precise and significant hypotheses, and do so extensively and fairly.

My recommendation is to accept this paper.

References:

[1] Shchur et al. Pitfalls of graph neural network evaluation. Neurips R2L 2018

[2] Errica et al. A fair comparison of graph neural networks for graph classification. ICLR 2020

[3] Lv et al. Are we really making much progress?: Revisiting, benchmarking and refining heterogeneous graph neural networks. KDD 2021

[4] Luo et al. Classic GNNs are strong baselines: Reassessing GNNs for node classification. Neurips 2024